peer-led interventions; low- and middle-income countries; mental Health; task-sharing

**Corresponding author:**
Dana Chow;
Email: danachow@u.duke.nus.edu

# A scoping review on peer-led interventions to improve youth mental health in low- and middle-income countries

Dana Chow[1] ⓘ, Dunstan J. Matungwa[2], Elizabeth R. Blackwood[3], Paul Pronyk[4,5] and Dorothy Dow[6,7,8]

[1]Duke-NUS Medical School, Singapore; [2]National Institute for Medical Research (NIMR), Mwanza, Tanzania; [3]Duke University Medical Center Library & Archives, School of Medicine, Durham, NC, USA; [4]Centre for Outbreak Preparedness, Duke-NUS Medical School, Singapore; [5]SingHealth Duke-NUS Global Health Institute, Singapore; [6]Duke Global Health Institute, Durham, NC, USA; [7]Kilimanjaro Christian Medical Center-Duke Collaboration, Moshi, Tanzania and [8]Department of Pediatrics, Infectious Diseases, Duke University Medical Center, Durham, NC, USA

## Abstract

Youth living in low- and middle-income countries (LMICs) have an increased vulnerability to mental illnesses, with many lacking access to adequate treatment. There has been a growing body of interventions using task sharing with trained peer leaders to address this mental health gap. This scoping review examines the characteristics, effectiveness, components of peer delivery and challenges of peer-led mental health interventions for youth aged 10–24 in LMICs. A key term search strategy was employed across MEDLINE, Embase, Web of Science, Global Health and Global Index Medicus. Eligibility criteria included young people aged 10–24 and a peer-led component delivered in any setting in an LMIC. Study selection and extraction were conducted independently by the first and second authors, with discrepancies resolved by the senior author. Study characteristics were summarised and presented descriptively. The search identified 5,358 citations, and 19 studies were included. There were 14 quantitative, four qualitative and one mixed methods study reporting mental health outcomes. Types of interventions were heterogenous but fell within three broad categories: (1) peer education and psychoeducation, (2) peer-led psychotherapy and counselling and (3) peer support. All studies reported improved mental health outcomes as a result of the peer-led interventions. Peer-led interventions are versatile in terms of both the types of interventions and mode of delivery. Lived experience, mutual respect and reduced stigma make this method a highly unique and effective way to engage this age group. However, implementing peer-led youth interventions is not without challenges. Adequate training, supervision, cultural appropriateness and support from established institutions are critical to safeguarding and ensuring the sustainability of such programs. Our findings suggest that peer-led models are a valuable intervention strategy that policymakers can leverage in current and future efforts to address youth mental health in LMICs. Future areas of research should expand to include the perspectives of other key stakeholders involved in the implementation of peer-led mental health interventions, focusing on factors including fidelity, feasibility and acceptability to enhance implementation insights.

## Impact statement

Adolescence, a critical stage marked by significant physical, mental, and social changes, often leaves youth vulnerable to mental health challenges. This vulnerability is exacerbated in low- and middle-income countries (LMICs), where there is an increased risk of mental distress due to poverty, environmental instability, and limited access to healthcare services. In LMICs, where 90% of the world's youth reside, the mental health treatment gap is stark. Task-shifting, which involves redistributing mental health care tasks to less specialised workers, has emerged as a viable solution to address this gap. This scoping review examines the effectiveness of peer-led mental health interventions for youth aged 10–24 in LMICs, highlighting their potential to improve mental health outcomes through relatable, culturally sensitive support. Despite diverse approaches – ranging from psychoeducation and psychotherapy to peer support – the consistent positive impact underscores the potential of peer-led strategies. The success of such interventions hinges on rigorous training, ongoing supervision, cultural sensitivity, and institutional backing to ensure their effectiveness and sustainability. This review highlights the versatility, impacts and challenges of peer-led interventions, advocating for their broader implementation to enhance youth mental health in LMICs.



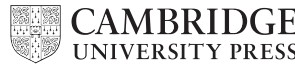

## Introduction

Adolescence is a stage of life marked by pronounced changes to the body, mind, and social environment (Lowenthal et al. 2014). The intricacies of navigating through this turbulent period often leaves youth vulnerable to mental distress. Approximately 75% of mental health disorders present by the age of 24 years (Das et al. 2016). Youth living in low- or middle-income countries (LMICs) are at an increased vulnerability to mental illnesses due to circumstances such as increased prevalence of poverty, environmental instability, and lack of access to medical and mental health services (Pedersen et al. 2019). In LMICs, where 90% of the world's youth live, the mental health treatment gap is the most severe (United Nations 2017). The median number of psychiatrists per 100,000 population in LMICs is 0.05, compared to 8.59 in high-income countries (HICs), resulting in three-quarters of individuals in need of mental health services unable to access them (Luitel et al. 2015; Le et al. 2022).

Task-shifting or sharing has been increasingly cited as a potential approach to respond to unmet needs of countries with a scarcity of specialised mental health care providers (Triece et al. 2022). Task-shifting or sharing is the redistribution of care usually provided by those with formalised training (i.e. psychiatrists) to individuals with little or no formal training (i.e. lay health workers). The literature has shown that with training and continued supervision, lay health workers can effectively deliver psychological treatments (van Ginneken et al. 2013).

Specifically, there has been a growing body of studies using task sharing with peer groups for intervention delivery (Stokar et al. 2014). A peer is defined as a person who shares common sociodemographic characteristics and/or lived experiences of a condition, event, or disorder with the target population (Farkas and Boevink 2018; Pedersen et al. 2019). In the literature, trained peers are referred to with various terms, including "peer-leaders" (Stokar et al. 2014), "peer educators," "peer-volunteers" (Maselko et al. 2020), "peer-facilitators" (Alcock et al. 2009), "peer-counsellors" (Nankunda et al. 2010), "peer-mentors" (Shroufi et al. 2013) and "peer-supporters" (Nkonki et al. 2010). For the purpose of this review, we will use the term "peer leaders" to refer to trained peers who deliver interventions. This will also distinguish these interventions from those involving peer support within group therapy sessions led by professional mental health specialists. Peer-led mental health services are grounded on the concept of peer support, which is the provision of emotional appraisal and informational assistance through access to a social network (Dennis 2003; Embuldeniya et al. 2013). The knowledge of a disease stems from lived experiences and peer leaders are typically trained to deliver specific interventions but lack professional status through academic or professional qualifications (Pedersen et al. 2019). This camaraderie provides qualities unique to peer support by offering validation of lived experiences, building rapport, and establishing bonds (Mancini 2018; Linnemayr et al. 2021). Individuals who are from the local community are found to relate better to the target group and could share approaches to dealing with distress that were in line with existing sociocultural contexts (Petersen et al. 2012). This was particularly useful for cultural nuances, that may be delicate for an outsider to address, such as prevailing customs or patriarchal realities. Peer delivery of mental health interventions has also been proposed as a cost-effective yet feasible and acceptable solution to address human resource challenges (Sikander et al. 2019; Maselko et al. 2020) thereby filling a gap in mental health treatment.

There have been several reviews of peer-led mental health interventions, but none specifically explored their impact on youth mental health in LMICs – a unique demographic with a specific set of challenges – and led by peers who are themselves youth (Sikkema et al. 2015; Hoeft et al. 2018; Bhana et al. 2021; de Beer et al. 2022). This scoping review profiles the evidence for peer-led interventions to improve the mental health of youth aged 10–24 and living in LMICs with the aim of informing the development of impactful interventions to address this crucial gap. Key research questions we explored include:

1. What are the characteristics and outcomes of existing peer-led mental health interventions in LMICs?
2. What are the components required to enable peer-leaders to act as key agents in intervention delivery?
3. How effective are peer-led interventions in improving youth mental health outcomes in LMICs? What mechanisms make them more effective than if led by adults or non-peers?
4. What are the challenges to implementing such interventions?

## Methods

### Search strategy

The search was developed and conducted by a professional medical librarian in consultation with the author team and included a mix of keywords and subject headings representing adolescents, mental health, peer support, and LMICs. The searches were independently peer-reviewed by another librarian using a modified PRESS Checklist (McGowan et al. 2016).

Search hedges or databases filters were used to remove publication types such as editorials, letters, books, book chapters, essays, conference proceedings, and comments as was appropriate for each database. Searches were conducted in MEDLINE via Ovid, Embase via Elsevier, Web of Science via Clarivate, Global Health via EBSCO, and Global Index Medicus via WHO. The searches were executed on 31 August 2023 and found 3,404 unique citations. A search update conducted on 6 September 2024 to identify newly published studies that found an additional 403 studies. Key search terms included "adolescent," "youth," "mental health," "peer" and "LMIC." Complete reproducible search strategies, including search filters, for all databases are detailed in the Supplementary Materials.

### Study selection

All citations were imported into Covidence, a systematic review screening software, which was also used to remove duplicates. Initial inclusion and exclusion based on the title and abstracts were made independently by the first author (DC) and second author (DJM). Potentially relevant studies were selected and the full text reviewed by the first and second authors. Any discrepancies were resolved by the senior author (DD).

### Eligibility criteria

Eligibility criteria were specified based on the PICOS model of Population, Intervention, Comparison, Outcome and Study type. Included studies were required to have an intervention that included young people aged 10–24 making up at least 50% of the study population, a peer-led component, has been conducted in any setting (i.e. community, hospital, schools, refugee camps, online

etc.), and been delivered in a LMIC. LMIC was defined according to the World Bank List. While we recognise the significant diversity among LMICs, spanning various continents and cultural contexts, we have chosen to include all LMICs as they share common structural and socioeconomic barriers that impact the delivery of health interventions. By including a broad spectrum of LMICs, this study can identify patterns and insights that can inform scalable, context-specific solutions. Both quantitative and qualitative data were acceptable but had to include a measured form of mental health, either distress (i.e. depression, anxiety, suicidal ideation attempt etc.) or wellness (i.e. resilience, self-efficacy etc.). To be included, qualitative outcomes had to specifically document a change in mental health/distress in relation to a peer-led intervention.

### Data extraction

Data extraction was conducted using Covidence to support the extraction of relevant and detailed information including author, year of publication, study design, country, and intervention details. The extraction was similarly performed by authors DC and DJM and discrepancies resolved by DD.

### Data analysis

The characteristics of the studies were summarised and presented descriptively to illustrate the scope of the included literature. Due to the heterogeneity of studies and scales used, it was not possible to conduct a meta-analysis. Qualitative studies were reviewed for any specific outcomes pertaining to mental health, with relevant quotations from participants extracted from the articles.

### Results

#### Characteristics of included studies

The search identified 5,358 citations, and after excluding duplicates, 3,807 articles were screened using title and abstract. We excluded 3,752 studies for key reasons such as irrelevance to the topic, incorrect target population, intervention and study type. We manually added three studies (Simms et al. 2022; Venturo-Conerly et al. 2022b, 2024). Fifty-eight articles underwent full text screening based on criteria described, and 19 were included (Balaji et al. 2011; Im et al. 2018; Mathias et al. 2018, 2019; Yuksel and Bahadir-Yilmaz 2019; Dow et al. 2020; Osborn et al. 2020a, 2021, 2023; Duby et al. 2021; Kermode et al. 2021; Mohamadi et al. 2021; Simms et al. 2022; Venturo-Conerly et al. 2022b, 2024; Ferris France et al. 2023; Harrison et al. 2023; Merrill et al. 2023; Tinago et al. 2024) (Figure 1, Table 1). Included studies were conducted between 2011 and 2024. Six studies were conducted in Asia-Pacific: four in India (Balaji et al. 2011; Mathias et al. 2018; Mathias et al. 2019; Kermode et al. 2021), one in Iran (Mohamadi et al. 2021), one in Turkey (Yuksel and Bahadir-Yilmaz 2019). The rest were based in Africa: one in Tanzania (Dow et al. 2020), two in South Africa (Duby et al. 2021; Harrison et al. 2023), three in Zimbabwe (Simms et al. 2022; Ferris France et al. 2023; Tinago et al. 2024), one in Zambia (Merrill et al. 2023) and six in Kenya (Im et al. 2018; Osborn et al. 2020a, 2021; Venturo-Conerly et al. 2022b, 2024; Osborn et al. 2023). Seven were randomised controlled trials (RCTs) (Balaji et al. 2011; Dow et al. 2020; Osborn et al. 2020a; Osborn et al. 2021; Merrill et al. 2023; Osborn et al. 2023; Venturo-Conerly et al. 2024), three cluster-randomised (Mohamadi et al. 2021; Simms et al. 2022; Venturo-Conerly et al. 2022b) and the rest were quasi-experimental trials (Im et al. 2018; Mathias et al. 2018; Mathias et al. 2019; Yuksel and

Bahadir-Yilmaz 2019; Duby et al. 2021; Kermode et al. 2021; Ferris France et al. 2023; Harrison et al. 2023; Tinago et al. 2024). There were 14 quantitative (Balaji et al. 2011; Im et al. 2018; Mathias et al. 2018; Yuksel and Bahadir-Yilmaz 2019; Osborn et al. 2020a; Dow et al. 2020; Kermode et al. 2021; Mohamadi et al. 2021; Osborn et al. 2021; Simms et al. 2022; Venturo-Conerly et al. 2022b, 2024; Osborn et al. 2023; Tinago et al. 2024), four qualitative (Mathias et al. 2019; Duby et al. 2021; Ferris France et al. 2023; Merrill et al. 2023) and one mixed-methods (Harrison et al. 2023) study. Five studies were pilot trials (Balaji et al. 2011; Dow et al. 2020; Osborn et al. 2020a; Ferris France et al. 2023; Harrison et al. 2023). Mathias and (2018, 2019) and Kermode et al. (2021) were studies on a related intervention based on the Nae Disha Curriculum, while Osborn et al. (2021, 2020a) and Venturo-Conerly et al. (2022b, 2024) were studies on the Shamiri Intervention. The majority of studies focused on a general population of youth experiencing mental health difficulties, but four had specific focus on youth living with human immunodeficiency virus (HIV) (Dow et al. 2020; Simms et al. 2022; Ferris France et al. 2023; Merrill et al. 2023), four on girls and young women (Mathias et al. 2018; Duby et al. 2021; Mohamadi et al. 2021), including one on adolescent mothers (Tinago et al. 2024), and one on youth refugees (Im et al. 2018).

#### Types of interventions

The studies were heterogenous in terms of the types of interventions, and fell into three main categories: (1) peer-led psychotherapy and counselling: a therapeutic process facilitated by trained peers, focusing on improving participants' mental health through structured therapy or counselling sessions, (2) peer education and psychoeducation: an approach where peer leaders teach participants about specific health or psychological topics, with the goal to equip individuals with knowledge and techniques to enhance their mental health and well-being, and (3) peer support: a system of informal, mutual support, where peers provide social connection, encouragement, and a sense of community to foster mental health and well-being. Peer-led psychotherapy and counselling, which require the highest level of specialisation, often include elements of peer education. All three types of interventions integrate components of peer support. Eight studies (Dow et al. 2020; Osborn et al. 2020a; Osborn et al. 2021; Venturo-Conerly et al. 2022b, 2024; Ferris France et al. 2023; Osborn et al. 2023) incorporated psychotherapy in their intervention, delivered in settings including the community, online, schools, and health clinics. Psychological therapies included trauma-informed cognitive behavioural therapy (TI-CBT), interpersonal psychotherapy (IPT), motivational interviewing (MI), meditation, inquiry-based stress reduction (IBSR), an art-literacy program, and problem solving therapy (PST). All were conducted in a group setting, but Dow et al (2020) and Ferris France et al (2023) had also weaved in individual sessions or activity components. Eight studies (Balaji et al. 2011; Im et al. 2018; Mathias et al. 2018; Mathias et al. 2019; Yuksel and Bahadir-Yilmaz 2019; Kermode et al. 2021; Mohamadi et al. 2021; Merrill et al. 2023) used peer education and psychoeducation in their interventions. Settings included schools, universities, communities, and an HIV clinic. Im et al. (2018) explored the Trauma-Informed Psychoeducation (TIPE) intervention to promote refugee resilience, conflict resolution, impact of trauma on the body and emotional coping. Kermode et al. (2021) and Mathias et al. (2018, 2019) ran interventions adapted from the same Nae Disha curriculum involving the formation of Youth Wellness Groups and interactive modules on self-identity, self-esteem, relationship and communication

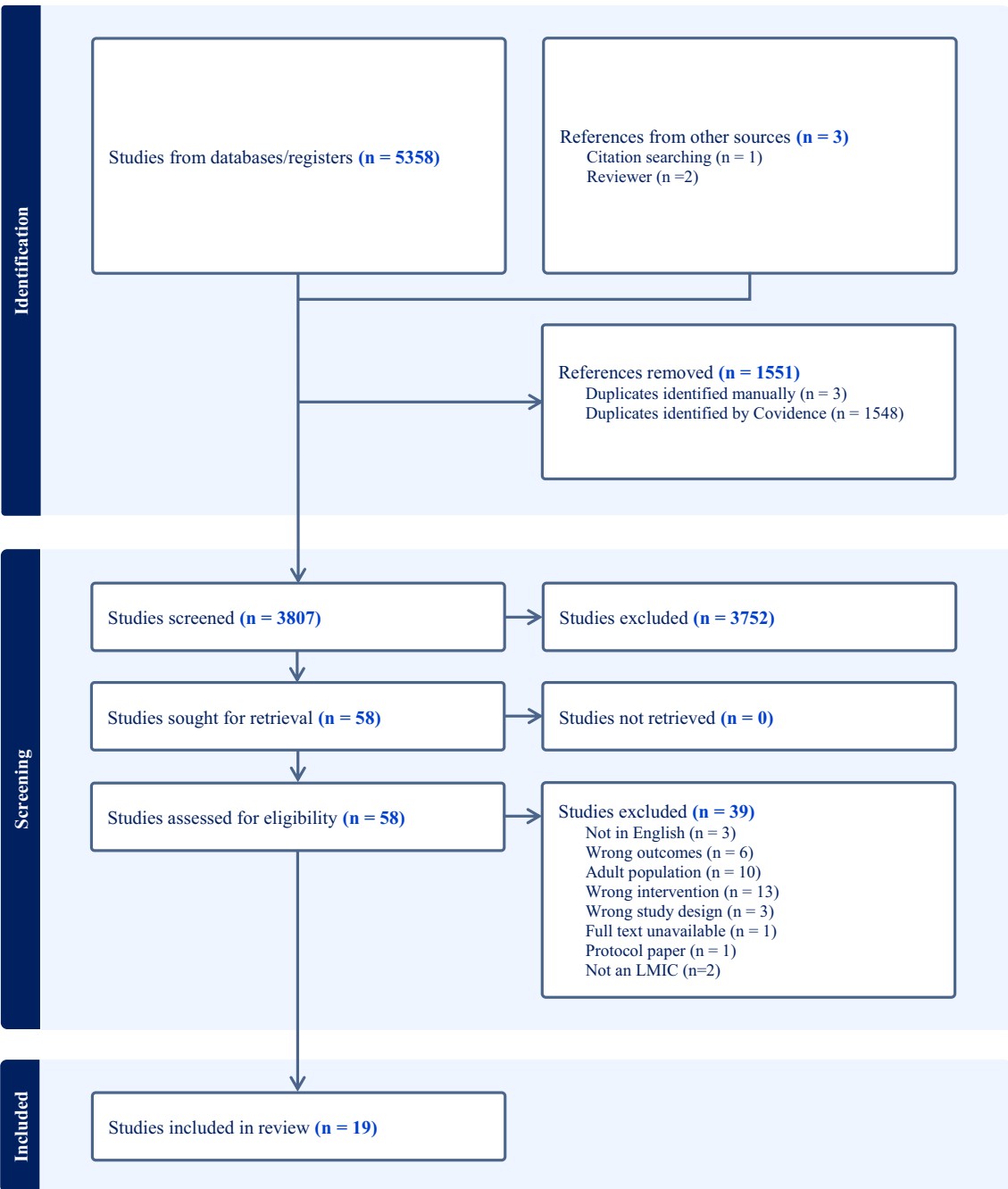

**Figure 1.** PRISMA flowchart.

skills, drawing boundaries, and managing emotions. Most were delivered as classroom-based group sessions, with Merrill et al. (2023) additionally incorporating individual meetings and Balaji et al. (2011) organising street plays. Peer support was utilised by three studies (Duby et al. 2021; Harrison et al. 2023; Tinago et al. 2024), and involved the creation of spaces to form social networks and facilitate discussions at hospitals, universities, and community clubs.

### Description of peer-leaders: Recruitment, training and support

A variety of terms were used to refer to the peer leaders, including "peer-educators," "community adolescent treatment supporters,"
"community youth leaders," "group leaders," "youth peer mentors," "student lay supporters," "peer mentors," "lay-providers." All studies involved peer leaders who were locally based lay-persons, similar in age group and shared geographical, linguistic, cultural and in most cases, health conditions with the target population, along with no formal qualification, qualifying them to be called peers. The age group of peer leaders ranged from 18 to 30 years old, although several studies did not specify it (Table 2).

The criteria to be a peer leaders varied according to the target community but shared similarities in terms of language and skills requirements. Four studies (Dow et al. 2020; Simms et al. 2022; Ferris France et al. 2023; Merrill et al. 2023) with a focus on young people living with HIV (YPLWH) recruited peer leaders who were

**Table 1.** Data extraction of included studies

| Author, year and country | Aim, study design and data type | Participants | Intervention and setting | Control or comparator | Description of peer leaders | Mental health measure | Key findings |
|---|---|---|---|---|---|---|---|
| **Psychotherapy and counselling** | | | | | | | |
| Dow et al. 2020 Country: Tanzania | Aim: To establish (1) the feasibility and acceptability of the pilot Sauti ya Vijana (SYV) intervention and (2) to conduct exploratory analyses on the impact of SYV on outcomes including mental health, stigma, ART adherence and HIV RNA Study design: Randomised controlled trial Data type: Quantitative | Young people living with HIV who attended the adolescent HIV clinic at Kilimanjaro Christian Medical Centre (KCMC) or Mawenzi Regional Referral Hospital (MRRH) Age range: 12–24 Mean age: 18.1 N = 93 | Intervention: Psychotherapy Description: SYV, 10 weekly group sessions (two held jointly with caregivers) and two individual sessions incorporating trauma informed cognitive behavioural therapy, interpersonal psychotherapy, and motivational interviewing. Participants were split into same sex groups with sex-concordant peer leaders. Setting: Community | Routine monthly attendance to adolescent clinic with adherence counselling following Tanzanian guidelines | Young adults (23–30 years old) with background of either living with HIV and/or having prior experience with mental health research. They underwent a two-week intensive training with the Principal Investigator and clinical psychologists with weekly supervision sessions. | • Patient Health Questionnaire (PHQ–9) • Strengths and Difficulties Questionnaire (SDQ) • UCLA Post Traumatic Stress Symptoms Exposure Screener and Reaction Index Survey | • Mental health scores had greater improvements in the intervention arm compared to the standard-of-care (SOC) arm • Change in PHQ–9 score − 0.60 (95% CI −2.67, 1.47) • Change in SDQ Total Difficulties Score 0.88 (95% CI −3.22, 1.47) • Change in UCLA Trauma Score − 0.03 (95% CI −2.38, 2.32) |
| Ferris France et al. 2023 Country: Zimbabwe | Aim: To increase self-worth and wellbeing by reducing self-stigma among adolescents and young people living with HIV (AYPLHIV) Study design: Quasi-experimental design Data type: Qualitative | Zvandiri Community Adolescent Treatment Supporters (CATS) living with HIV (LHIV), trained, mentored and supported to deliver structural support groups, counselling and tailored community-based adherence support to their peers. Age range: 18–24 N = 60 | Intervention: Psychotherapy Description: The Wakakosha intervention (second delivery by peer coaches), a 10-day face-to-face intervention consisting of theory, meditation, group and individual experiences of Inquiry-Based Stress Reduction (IBSR), music reflection, sharing of insights, singing and songs, activity journaling, poem and letter writing, and podcast making Setting: Community and online (COVID–19) | None | 15 CATS identified from the first delivery of the intervention were trained as peer coaches to lead the second delivery of the intervention. They attended residential face-to-face, six-day immersive Training of Trainers. They were supported by adult coaches when delivering the intervention | • Self-confidence and self-agency • Sense of purpose and meaning in life • Self-forgiveness and forgiveness of others | • Participants gained various skills including self-confidence • Participants affirmed a new sense of purpose in their lives and inspired a sense of self-worth • Participants experienced positive changes in self-forgiveness and forgiveness of parents for passing on HIV |
| Osborn et al. 2020a Country: Kenya | Aim: To evaluate if students in the Shamiri intervention would experience reductions in depressive and anxiety symptoms and improvements in social support, perceived control, and academic outcomes in a preliminary proof-of-concept trial. Study design: Randomised controlled trial Data type: Quantitative | Adolescents with depression or anxiety Age range: 12–19 N = 51 | Intervention: Psychotherapy Description: The Shamiri intervention consists of four weekly sessions that use didactic lectures, reading activities, group discussions and writing activities to explore self-growth, gratitude and values to address depression and anxiety Setting: School | Didactic sessions on study skill strategies, group discussions and activities. | Five trained group leaders (17–21 years old) who are high school graduates and able to read in English. Training included 20 h of session. | • PHQ–8 • GAD–7 • Multidimensional Scale of Perceived Social Support (MSPSS) • Perceived Control Scale for Children (PCS) | Depressive symptoms • Intervention: Youths reporting mod-severe-to-severe depressive symptoms decreased by 21.42% • Control: Youths reporting mod-severe-to-severe depressive symptoms decreased by 4.35% • At end point, only 21.44% of adolescents in intervention qualify for intervention on merit of depressive symptoms, compared to 26.09% in the study skills |

*(Continued)*

*Cambridge Prisms: Global Mental Health*

| Author, year and country | Aim, study design and data type | Participants | Intervention and setting | Control or comparator | Description of peer leaders | Mental health measure | Key findings |
|---|---|---|---|---|---|---|---|
| | | | | | | | Anxiety symptoms<br>• Anxiety symptoms declined more rapidly from baseline to Week 4 for youth in intervention. ($p = 0.039$, $d = .54$ [−.20,1.29]<br>• Intervention: Youths reporting mod-to-severe anxiety symptoms decreased by 42.86%<br>• Control: Youths reporting mod-to-severe anxiety symptoms decreased by 17.39%<br>• At end point, only 46.43% of adolescents in intervention qualify for intervention on merit of anxiety symptoms, compared to 78.26% in the study skills |
| Osborn et al. 2021 Country: Kenya | Aim: To assess whether Shamiri intervention can alleviate depression and anxiety symptoms in symptomatic Kenyan adolescents<br>Study design: Randomised controlled trial<br>Data type: Quantitative | Adolescents with elevated symptoms on standardised depression or anxiety measures<br>Age range: 13–18 years<br>Mean age: 15.4<br>$N = 413$ | Intervention: Psychotherapy<br>Description: The Shamiri intervention consists of four weekly sessions that use didactic lectures, reading activities, group discussions and writing activities to explore self-growth, gratitude and values to address depression and anxiety<br>Setting: School | Peer-led study skills session to teach skills on note-taking, study strategies, time management, study cycle. | High school graduates fluent in English and Kiswahili 18–26 years old who underwent 10 h of training and weekly supervision. | • PHQ–8<br>• GAD–7 | Depression<br>• 2-week follow-up: non-significant<br>• End-point: Youths in Shamiri experienced greater declines in depression symptoms from baseline to end point than control group youths (Cohen $d = 0.35$ [95% CI, 0.09–0.60]<br>Anxiety<br>• 2-week: Symptoms reduced significantly more for youths in Shamiri than in control<br>• 7-month: significantly lower<br>• Anxiety symptoms at 7-month follow-up for the Shamiri than study skills groups (imputed values model: $B = 1.78$ [95% CI, 0.53–3.03]; $t = 2.8$; $df = 25.54$; $p = .01$; unimputed values model: $B = 2.25$ [95% CI, 1.08–3.4]; $t = 3.8$; $df = 345.25$; $p < .001$) |

*Cambridge Prisms: Global Mental Health*

| Author, year and country | Aim, study design and data type | Participants | Intervention and setting | Control or comparator | Description of peer leaders | Mental health measure | Key findings |
|---|---|---|---|---|---|---|---|
| | | | | | | | • End-point: Youths in Shamiri experienced greater declines in anxiety symptoms from baseline to end point than youths in the control group per the model using unimputed values (Cohen $d$ = 0.37 [95% CI, 0.11–0.63] |
| Osborn et al., 2023 Country: Kenya | Aim: To evaluate the effectiveness of Pre-Texts arts-literacy intervention for adolescent depression and anxiety in Kenyan high school students Study type: Randomised Controlled Trial Data type: Quantitative | Students in Grades 9–11 from two community-run schools Age range: 12–19 Mean age: 16.36 $N$ = 235 (intervention = 106 and control = 129) | Intervention: Psychotherapy Description: An afterschool Pre-Texts art literacy intervention involving group (5–6 youth) meetings of 1 h over a one-week period for a total of five meetings. Literary, technical, scientific text extracts are provided and participants are invited to use the text as inspiration to understand and engage with themes and messages, thereby facilitating psychological change. This is followed by reflective discussions. Setting: School | Study skills control group involving notetaking, time management, effective reading strategies. | Trained youth facilitators with a high school diploma and from around the Nairobi area. | • PHQ–8 • GAD–7 | • Non-significant effects for time, intervention condition and covariates age and sex • Significant Time X Condition interaction suggests that youth who receive Pre-Text intervention experienced greater reductions in depressive symptoms from baseline to 1-month follow up compared to those in the control group. ($d$ = 0.52, 95% CI [0.19, 0.84]) • Significant Time and Time X Condition effect at 1 month follow up that those participating in the intervention experienced greater reductions in anxiety from baseline to 1-month follow up compared to control-group youth ($d$ = 0.51, 95% CI [0.20, 0.81]) • Adolescents who received Pre-Texts intervention experienced larger declines in depression symptoms from baseline to 1-month follow-up than control group youths ($d$ = 1.10, 95% CI [0.46, 1.75]) • Youth who received Pre-Texts generally experienced larger declines in anxiety symptoms from baseline to 1-month |

| Author, year and country | Aim, study design and data type | Participants | Intervention and setting | Control or comparator | Description of peer leaders | Mental health measure | Key findings |
|---|---|---|---|---|---|---|---|
| | | | | | | | follow-up than control group youths ($d$ = 0.54, 95% CI [−0.07, 1.45]) |
| Simms et al. 2022 Country: Zimbabwe | Aim: To evaluate if enhancing the counselling skills of CATS to provide problem-solving therapy (PST) reduces virological non suppression and improves mental health among adolescents living with HIV in Zimbabwe, compared with standard Zvandiri care Study design: Cluster-randomised trial Data type: Quantitative | Adolescents living with HIV Age range: 10–19 $N$ = 842 | Intervention: Psychotherapy Description: Zvandiri-Problem-solving therapy (Zvandiri-PST) which consists of standard Zvandiri program (HIV care, counselling and home visits) and PST (a cognitive behavioural approach for problem solving) Setting: Health clinic | Zvandiri standard care consisting of HIV care following Ministry of Health and Child Care guidelines, counselling and home visits by trained CATS, monthly support groups and weekly text messages | Community Adolescent Treatment Supporters (CATS) who are 18–24 years old living with HIV who are trained and mentored to provide peer counselling and support | • Shona Symptom Questionnaire (SSQ) • PHQ–9 | Intervention effect on prevalence and severity of common mental health outcomes • SSQ ≥ 8: 2.4% versus 10.3% (AOR = 0.19; 95% CI 0.08, 0.46; $p$ < 0.001; AMD = −1.14; 95% CI −1.80, −0.49; $p$ = 0.001) • PHQ–9 ≥ 10: 2.9% versus 8.8%; (AOR = 0.32; 95% CI 0.14, 0.78; $p$ = 0.01; AMD = −1.14; 95% CI −2.01, −0.27; $p$ = 0.01) |
| Venturo-Conerly et al., 2022b Country: Kenya | Aim: To test key, short term outcomes of a universal, classroom-based, single session version of each component of Shamiri intervention (i.e. growth, gratitude, and value affirmation) against an active control intervention. Study design: Cluster Randomised Trial Data type: Quantitative | Students from two sub-county (lowest academic ranking) public boarding high schools in Kiambu Country, Kenya, one an all-girls and one an all-boys school $N$ = 895 Mean age: 16.2 (all-girls school), 15.86 (all-boys school) | Intervention: Psychotherapy Description: A single 1-h session version of the Shamiri intervention (Osborn et al., 2020b) testing each component (i.e. growth, gratitude, value affirmation) separately Setting: School | Study skills control involving discussing helpful study strategies and implementation. | Lay-providers who were recent Kenyan high school graduates | • GAD–7 • PHQ–8 | • Value affirmation intervention reduced anxiety symptoms in the universal sample • Value affirmation ($B$ = −2.22, $p$ < .01; Cohen's $d$ = 0.49 [0.09–0.89]) and growth mindset ($B$ = −1.78, $p$ < .05; Cohen's $d$ = 0.39 [0.01–0.76]) interventions reduced anxiety in sub-sample endorsing moderate-to-severe symptoms at baseline • Adolescents in all conditions including control experienced decreases in self-reported depression symptoms from baseline to 2-week follow-up |
| Venturo-Conerly et al., 2024 Country: Kenya | Aim: To provide a more rigorous, long-term, and adequately powered examination of the Shamiri intervention, involving each intervention component, all | Students from four high-schools in Kenya $N$ = 1,252 Age range: 12–21 Mean age: 16.25 | Intervention: Psychotherapy Description: The Shamiri intervention consisting of all components (growth mindset, gratitude, and values affirmation) were tested against single-component sessions. Each intervention was tested | Study skills control involving discussing helpful study strategies and implementation | Lay-providers who were recent Kenyan high school graduates aged 18–22 | • GAD–7 • PHQ–8 • Short-Warwick-Edinburgh Mental Wellbeing Scale (SWEMWBS) | Within the main sample: • Anxiety scores significantly improved on average compared to baseline at two-week midpoint ($B$ = −0.847; [95%CI −1.57 −0.13]; $p.adj$ < 0.001), four-week endpoint |

| Author, year and country | Aim, study design and data type | Participants | Intervention and setting | Control or comparator | Description of peer leaders | Mental health measure | Key findings |
|---|---|---|---|---|---|---|---|
| | components combined, and the study-skills condition.<br>Study design: Randomised Controlled Trial<br>Data type: Quantitative | | against an active control group.<br>Setting: School | | | | ($B = -2.948$; [95%CI $-3.60$ $-2.30$]; *p.adj* < 0.001), one-month follow-up ($B = -1.587$; [95%CI $-2.55$ $-0.63$]; *p.adj* < 0.001), three-month follow-up ($B = -2.374$; [95%CI $-2.99$ $-1.76$]; *p.adj* < 0.001), and eight-month follow-up ($B = -1.917$; [95%CI $-2.59$ $-1.25$]; *p.adj* < 0.001).<br>• Depression scores were significantly improved in full sample compared to baseline at two-week midpoint ($B = -0.796$; [95%CI $-1.67$ 0.08]; *p.adj* = 0.011), four-week endpoint ($B = -3.126$; [95%CI $-3.79$ $-2.46$]; *p.adj* < 0.001), one-month follow-up ($B = -2.382$; [95%CI $-3.53$ $-1.23$]; *p.adj* < 0.001), three-month follow-up ($B = -2.521$; [95%CI $-3.42$ $-1.62$]; *p.adj* < 0.001), and eight-month follow-up ($B = -2.237$; [95%CI $-3.19$ $-1.29$]; *p.adj* < 0.001).<br>• Well-being scores were significantly improved than baseline at two-week midpoint ($B = 1.73$; [95%CI 0.76 2.66]; *p.adj* < 0.001), four-week endpoint ($B = 3.44$; [95%CI 2.27 4.60]; *p.adj* < 0.001), one-month follow-up ($B = 2.21$; [95%CI $-0.32$ 4.75]; *p.adj* = 0.02), three-month follow-up ($B = 1.78$; [95%CI 0.09 3.47]; *p.adj* = 0.004), and eight-month follow-up ($B = 1.59$; 16 [95%CI 0.35 2.84]; *p.adj* < 0.001).<br>• There were no significant differences between conditions on measures of |

*Cambridge Prisms: Global Mental Health*

| Author, year and country | Aim, study design and data type | Participants | Intervention and setting | Control or comparator | Description of peer leaders | Mental health measure | Key findings |
|---|---|---|---|---|---|---|---|
| | | | | | | | depression, anxiety, or wellbeing. Within the clinical sub-sample: <br> • Anxiety scores significantly improved compared to baseline at two-week midpoint ($B = -4.00$; [95%CI $-5.18$ $-2.82$]; $p.adj < 0.001$), four-week endpoint ($B = -7.13$; [95%CI $-8.13$ $-6.14$]; $p.adj < 0.001$), one-month follow-up ($B = -6.51$; [95%CI $-8.02$ $-4.99$]; $p.adj < 0.001$), three-month follow-up ($B = -6.26$; [95%CI $-7.54$ $-4.99$]; $p.adj < 0.001$), and eight-month follow-up ($B = -6.10$; [95%CI $-7.25$ $-4.95$]; $p.adj < 0.001$). <br> • Depression scores were significantly improved in full sample compared to baseline at two-week midpoint ($B = -2.89$; [95%CI $-4.01$ $-1.77$]; $p.adj < 0.001$), four-week endpoint ($B = -6.60$; [95%CI $-7.54$ $-5.66$]; $p.adj < 0.001$), one-month follow-up ($B = -6.34$; [95%CI $-7.57$ $-5.11$]; $p.adj < 0.001$), three-month follow-up ($B = -5.45$; [95%CI $-6.75$ $-4.14$]; $p.adj < 0.001$), and eight-month follow-up ($B = -5.29$; [95%CI $-6.59$ $-3.99$]; $p.adj < 0.001$). <br> • Symptoms of depression and anxiety showed no significant differences across five groups. <br> • Wellbeing scores significantly improved at two-week midpoint than at baseline ($B = 2.51$; [95%CI 1.39 3.64]; $p.adj < 0.001$), 4-week endpoint ($B = 5.02$; |

| Author, year and country | Aim, study design and data type | Participants | Intervention and setting | Control or comparator | Description of peer leaders | Mental health measure | Key findings |
|---|---|---|---|---|---|---|---|
| | | | | | | | [95%CI 3.02 7.02]; *p. adj* < 0.001), 1-month follow-up (*B* = 4.41; [95% CI 3.12 5.69]; *p. adj* < 0.001), three-month follow-up (*B* = 3.38; [95% CI 1.97 4.80]; *p. adj* < 0.001), and eight-month follow-up (*B* = 3.19; [95%CI 1.87 4.52]; *p.adj* < 0.001) |
| **Peer education and psychoeducation** | | | | | | | |
| Balaji et al. 2011 Country: India | Aim: To assess the acceptability, feasibility, and potential effectiveness of the pilot project Yuva Mitr in improving a range of priority health outcomes for youths 16–24 in urban and rural communities in Goa Study design: Randomised controlled trial Data type: Quantitative | Youth living in urban communities in the wards of Margao and rural communities from the catchment area of Balli Primary Health Centre Age range: 16–24 Mean age: 19 *N* = 3,663 (baseline) *N* = 3,562 (follow-up) | Intervention: Peer education Description: A 12-month intervention comprised of institution-based and community peer education involving group sessions and street plays, a teacher training program and dissemination of health information materials Setting: Community and school | Not specified | Selected based on a pre-determined criteria (non-specified), trained by psychologists and social workers, and supported by community advisory board trained teachers. | • General Health Questionnaire (GHQ–12) | **Rural:** <br>• Probable depression decreased in the intervention arm by 60.1%, where it increased by 1.1% in the comparison arm (*p* < 0.001).<br>• Probable depression odds ratio 0.33, 95% CI 0.23–0.48<br>• Community peer education was feasible and acceptable<br>**Urban:**<br>• Suicidal behaviour decreased by 65.1% in the intervention arm compared to a decrease of 11.4% in the comparison arm (*p* = 0.02)<br>• Suicidal behaviour odds ratio 0.38 95% CI 0.17–0.84<br>• Probable depression decreased by 38.8% in the intervention arm compared to a decrease of 2.1% in comparison (*p* = 0.001)<br>• Probable depression odds ratio 0.57 95% CI 0.41–0.79<br>• Significant increase in knowledge levels for the topic on mental health<br>• Peer education in educational institutions showed acceptability but limited feasibility |

(*Continued*)

| Author, year and country | Aim, study design and data type | Participants | Intervention and setting | Control or comparator | Description of peer leaders | Mental health measure | Key findings |
|---|---|---|---|---|---|---|---|
| Im et al. 2018 Country: Kenya | Aim: To explore the effect of a trauma-informed psychoeducation (TIPE) intervention on both mental health and psychosocial domains among Somali refugee youth affected by trauma Study design: Quasi-experimental design Data type: Quantitative | Somali youth refugees living in an urban business district affected by multiple refugee traumas Mean age: 20 N = 141 | Intervention: Psychoeducation Description: 12 sessions of Trauma-informed Psychoeducation (TIPE) over three months that include (1) psychoeducational modules to promote refugee resilience, peace education, conflict resolution, problem-solving methods, (2) education on the impacts of trauma on the body, mind, social relationships, spirituality, and (3) psychosocial competencies such as emotional coping, problem solving, community and support systems and conflict management skills Setting: Community | None | 25 youth leaders Week long training of trainer (TOT) training 10 youth leaders paired with five community health counsellors | • PTSD Check List – Civilian Version (PCL-C) | • TIPE intervention had positive impacts on PTSD symptoms and psycho-social factors: • Youths with no/low base-line PTSD symptoms had their post-TIPE symptom score increase from 27.42–34.48 ($t = -4.476$ $p = 0.000$) • Youths with high PTSD score reported lower PTSD symptoms post-TIPE from 50.09–31.93 ($t = 8.188$ $p = 0.000$) |
| Kermode et al. 2021 Country: India | Aim: To implement and evaluate an intervention to promote social inclusion for young people affected by mental illness Study design: Quasi-experimental design Data type: Quantitative | Young people affected by mental illness, including those experiencing a mental health problem and those caring for a family member with a mental health problem Mean age: 18.9 N = 142 | Intervention: Psychoeducation Description: A 4–6-month intervention consisting of the formation of 11 Youth Wellness Groups (peer-facilitated, participatory groups of young people divided by gender) guided by a series of interactive modules adapted from the *Nae Disha* curriculum. Setting: Community | None | Eight peer facilitators affected by mental illness, with at least Class 12 education, ability to travel to intervention sites and communication skills. They were trained by the Burans Community Mental Health Project Team. (Qualitative outcomes of the same study has been described in Mathias et al., 2019) | • GHQ–12 • SDQ | • GHQ improved from 6.6 to 2.2 ($p < 0.001$) • SDQ improved from 16.1 to 11.8 ($p < 0.001$) |
| Mathias et al. 2018 Country: India | Aim: To evaluate the effectiveness of Nae Disha intervention to reduce anxiety and depression, promote attitudes of gender equality, self-efficacy and resilience among highly disadvantaged young women outside of a school setting in India Study design: Quasi-experimental repeated measures design Data type: Quantitative | Young women not attending school Age range: 12–24 Mean age: 16.7 N = 106 | Intervention: Psychoeducation Description: Nae Disha, a 15-week intervention to facilitate health promotion, development of psycho-social assets to promote youth citizenship and positive youth development. Modules include exploring self-identity and esteem, identifying and managing emotions, mental health communication skills, relationship skills and forgiveness, self-care and drawing boundaries Setting: Homes of community members | None | Local young women 20–30 years old expressing enthusiasm to work in youth resilience and have completed 12th Class. Had five days of training and two days of refresher training. | • Connor-Davidson Resilience Scale (CD-RISC) • Schwarzer's General Self-Efficacy Scale • PHQ–9 • GAD–7 | • Statistically significant improvement between pre- and post-intervention in all scales for self-efficacy, resilience, anxiety, depression and gender attitudes • Improvement in mental health and gender attitudes at follow up, but not in emotional resilience and self-efficacy. |

*Cambridge Prisms: Global Mental Health*

| Author, year and country | Aim, study design and data type | Participants | Intervention and setting | Control or comparator | Description of peer leaders | Mental health measure | Key findings |
|---|---|---|---|---|---|---|---|
| Mathias et al., 2019 Country: India | Aim: To assess the impact of a peer-led, community-based, participatory group intervention on social inclusion and mental health among young people affected by psycho-social disability in Dehradun district. Study type: Quasi-experimental design Data type: Qualitative | Young people affected by psycho-social disability (PSD) in four communities in Uttarakhand, India. Age range: 12–24 N = 142 | Intervention: Psychoeducation Description: The Nae Disha curriculum that builds on youth development and mental health promotion approaches using an interactive, participatory facilitation style of 17 learning modules (i.e. accepting differences, managing emotions, communicating confidently etc.). It also includes participation in community activities, access mental health services and visit a de-addiction centre. Setting: Community | None | Peer facilitators selected from the four target communities that were young people aged under 30 years, with personal experience of mental ill-health and who had completed 12th class in high school (Quantitative outcomes of the same study have been reported in Kermode et al., 2021) | • Intermediate and primary out-comes of the intervention • Mechanisms through which they were achieved | Experienced by both genders • Formation of new peer friendship networks • Increased self-efficacy • Improved mental health • Increased community participation Experienced by young women: • Increased freedom of movement • Greater confidence in communicating Experienced by young men: • Changes in community perceptions of them |
| Merrill et al., 2023 Country: Zambia | Aim: To explore youths' experiences with Project YES! to strengthen the understanding of the intervention's effectiveness and implementation, while enhancing the literature on peer-centered approaches to improving HIV outcomes among youth living with HIV Study design: Randomised Controlled Trial Data type: Qualitative | Youth living with HIV Age range: 15–24 N = 276 | Intervention: Peer education Description: Six-month peer-mentoring program, including an orientation meeting, and monthly individual and monthly group meetings with a youth peer mentor. The goal is to support youth to successfully adhere to ART and decreased internalised stigma to achieve viral suppression. Setting: HIV clinics | Standard of care, including regular clinic visits and option of joining monthly youth group meetings. | Youth Peer Mentors (YPM), aged 21–26 years old selected by healthcare providers as successfully managing their HIV. Completed a two-week training and underwent one month of practice meetings with youth before the intervention launch. | • Overcoming shame • Self-worth • Community building | • Participants overcame feelings of shame and developed greater self-worth. • Participants reported how community building could alleviate feelings of isolation |
| Mohamadi et al. 2021 Country: Iran | Aim: To compare the methods of motivational interviewing led by a specialist and peer-to-peer education in promoting the knowledge and performance about puberty health and mental health in adolescent girls Study design: Cluster randomised controlled trial Data type: Quantitative | Eighth grade adolescent female students Age range: 13–15 Mean age: 14.44 N = 334 | Intervention: Peer education Description: Peer educators led one formal training on puberty and mental health to other students, followed by the informal passage of information to peers in groups of 5–6 students within 1 month Setting: School | Two training sessions on puberty health conducted by the researcher Comparator: Group counselling involving motivational interviewing presented by a master in consultation in midwifery during five sessions of 60–90 min. | Active volunteers who scored higher on the puberty health questionnaire prior to the start of the study, responsible for transmitting information to 5–6 other students. | • Persian Standard Symptom Check-list–25 (SCL–25) | • Immediately and 1 month after intervention, the Intervention group scored significantly higher in knowledge and perform-ance, and improved in mental health than the control group • Scores in knowledge, performance, and mental health were better in the peer group (intervention 2) than intervention 1 and control group |

| Author, year and country | Aim, study design and data type | Participants | Intervention and setting | Control or comparator | Description of peer leaders | Mental health measure | Key findings |
|---|---|---|---|---|---|---|---|
| Yuksel et al. 2019 Country: Turkey | Aim: To determine the effect of mentoring program on adjustment to university and ways of coping with stress in nursing students Study design: Quasi-experimental study Data type: Quantitative | First-year undergraduate nursing students Mean age: N = 91 | Intervention: Peer education Description: Eight weekly sessions of peer mentoring program including acquaintance and group awareness, life in Aksaray and its features, communication skills, techniques that facilitate communication, interpersonal communication, stress and coping Setting: University | None | Ten fourth year students with 10 h of training over 5 days | • Ways of Coping Inventory (WCI) | • Posttest mean scores of active ways of coping with stress (Optimistic Approach and Seeking Social Support Approach) of the experimental group were statistically higher than that of the control group. |
| **Peer support** | | | | | | | |
| Duby et al. 2021 Country: South Africa | Aim: To explore the perceived benefits of participation in peer-group clubs to better understand how combination interventions can incorporate social support and mental health components that ensure their relevance and effectiveness Study design: Quasi-experimental design Data type: Qualitative | Adolescent girls and young women (AGYW) in and out of school Age range: 15–24 N = 185 | Intervention: Peer-support Description: Peer-group clubs (The Keeping Girls in School Program and Rise Young Women's Clubs) designed to build self-esteem, confidence, life skills, provide supportive peer networks, and offer a platform for group discussions on sexual and reproductive health rights and gender equality Setting: Community clubs and schools | None | Trained peer-educators of similar age | • Self-esteem • Empowerment, self-worth, and self-respect • Well-being and coping | Peer-group clubs: • Positively affected their well-being through building self-esteem and self-confidence • Allowed AGYW to feel empowered with improved self-worth and emotional strength • Improved mental health and wellness by teaching how to communicate feelings and emotions • Peer facilitation was regarded as beneficial to AGYW in providing SRH, emotional support and counselling |
| Harrison et al., 2023 Country: South Africa | Aim: A pilot study to examine the acceptability, feasibility and preliminary impact of a peer support group for youth living with a range of chronic illnesses Study type: Quasi-experimental design Data type: Mixed methods | Adolescents living with a range of chronic illness (i.e. HIV, renal disease, psychiatric conditions, diabetes, etc.) and receiving care at Groote Schuur Hospital (GSH) Age range: 13–24 Mean age: 18.74 N = 58 (intervention = 20 and control = 38) | Intervention: Peer-support Description: The Better Together Programme, which helps adolescents with chronic conditions build social networks that enhance psychosocial support, develop a sense of belonging with peers, create a space where adolescents can share their experiences and build empathy. Setting: Hospital | Non-peer group | Peer mentors who are also living with a chronic condition | Quantitative: • CD-RISC 10 • Child Attitude Toward Illness Scale (CATIS) • Beck Youth Inventories Second Edition (BYI-II) Qualitative: • Support and acceptance in support group • Benefits of participation in peer support group | Quantitative: • Those who attended at least five peer group sessions had higher self-reported individual-level resilience ($p = 0.004$), positive attitude toward their chronic illness ($p < 0.001$), stronger self-concept ($p = 0.039$), lower depressive symptoms ($p < 0.10$) • Average total or T-scores on Beck Anxiety Inventory for Youth, Beck Anger Inventory for Youth, and Beck Disruptive Inventory for Youth did not |

| Author, year and country | Aim, study design and data type | Participants | Intervention and setting | Control or comparator | Description of peer leaders | Mental health measure | Key findings |
|---|---|---|---|---|---|---|---|
| | | | | | | | statistically significantly differ between peer group and non-peer group patients.<br>• Statistically significant association between mental health improvement as a function of peer group participation overall (Wilks' Lambda = 0.7349, $F$ (8,49) = 2.21, p = 0.043) and for individual domains of resilience, attitudes toward illness, chronic disease stigma, self-concept, depression<br>• Attending more than 5 group sessions was positively associated with a reduced odds of screening positive for depression or anxiety<br>Qualitative<br>• An eye opening and powerful experience for most young people<br>• Finding support and acceptance in the support group<br>• Social support |
| Tinago et al., 2023 Country: Zimbabwe | Aim: To test the effectiveness of a community-based peer support intervention to mitigate social isolation and stigma of adolescent motherhood<br>Study type: Quasi-experimental design<br>Data type: Quantitative | Adolescent mothers who were pregnant and/or had a child or children who resided in the two selected communities in Harare<br>Age range: 15–18 years<br>$N$ = 104 (intervention), 79 (control) | Intervention: Peer-support<br>Description: Peer support groups that met in-person twice a month and completed 12 total 75-min peer-group sessions. WhatsApp was used to schedule meetings, answer queries or further discuss peer support topics. Average of 12 participants per group.<br>Setting: Community | A community similarly low-income and high-density in Harare. | 12 peer educators co-facilitated the peer support groups with Community Health Workers | • PHQ–9<br>• MSPSS<br>• SSQ<br>• Peer and Significant Adult Support (PSAS) | • PHQ–9 and SSQ had greater improvements in the intervention arm than control arm by 5.01 ($p$ < 0.001) and 3.10 ($p$ < 0.001) points<br>• Probability of moderate to severe depression decreased in intervention arm from 0.408 to 0.231 ($p$ = 0.005), and increased in control arm 0.267 to 0.333. Participants were four times less likely to become moderately-to-severely depressed than were control participants<br>• Each of the four scores of MSPSS (family, friends, significant other, and total) had greater |

(Continued)

**Table 1.** *(Continued)*

| Author, year and country | Aim, study design and data type | Participants | Intervention and setting | Control or comparator | Description of peer leaders | Mental health measure | Key findings |
|---|---|---|---|---|---|---|---|
| | | | | | | | improvements in the intervention arm than control by 0.691 ($p = 0.003$), 1.03 ($p < 0.001$), 1.61 ($p < 0.001$) and 1.11 ($p > 0.001$) points respectively<br>• Overall score of PSAS survey had greater improvement in intervention arm than control arm by 0.508 points ($p < 0.001$) |

also living with HIV. Dow et al. (2020) expanded their recruitment to also include young adults with prior experience with mental health research. At least 10 studies specified requirements for standard of education (Mathias et al. 2018; Mathias et al. 2019; Yuksel and Bahadir-Yilmaz 2019; Osborn et al. 2020a; Kermode et al. 2021; Osborn et al. 2021; Venturo-Conerly et al. 2022b, 2024; Osborn et al. 2023; Tinago et al. 2024) of which several also specified criteria for language (Osborn et al. 2020a; Osborn et al. 2021), leadership and communication skills, past experiences and interests (Yuksel and Bahadir-Yilmaz 2019; Kermode et al. 2021; Venturo-Conerly et al. 2022b, 2024; Osborn et al. 2023). Selection and recruitment processes for peer leaders were sometimes detailed. Osborn et al. (2023, 2021, 2020a) recruited from local universities and high school graduate forums using a written application and interview process. Venturo-Conerly et al. (2022b, 2024) selected youths based on online applications and in-person semi-structured interviews. Mathias et al. (2019) invited potential peer leaders for community meetings and interviews. Tinago et al. (2024) recruited peer educators via snowball sampling and in-person meetings. A few studies selected peer leaders instead, based on scores on a pubertal health questionnaire (Mohamadi et al. 2021), assessment of readiness and capacity (Simms et al. 2022), academic grade and interviews with faculty (Yuksel and Bahadir-Yilmaz 2019).

Within the psychotherapy intervention type, the training period spanned from 4 days to 3 weeks. Training topics included specific therapy techniques, interpersonal and leadership skills, and often utilised role playing. Notably, Osborn et al. (2023, 2020a), Simms et al. (2022), Venturo-Conerly et al. (2024) trained their peer leaders in clinical risk assessment and making referrals to relevant support (i.e. mental health services, school resources). Studies in the peer education and psychoeducation intervention type had their trainings spanning 1 day to 2 weeks on program content and curriculum. Notably, Merrill et al. (2023) held practice meetings for 1 month after the initial 2 week training period, and Mathias et al. (2018, 2019) had instated refresher training during the intervention period. Within the three studies utilising peer support as their intervention, only Tinago et al. (2024) had specified their training type, with a 3 day session on the subject matter. Most trainings were led by the research team. Four studies included a psychologist or mental health specialist (Balaji et al. 2011; Dow et al. 2020; Harrison et al. 2023; Venturo-Conerly et al. 2024), and Ferris France et al. (2023) used local adult coaches living with HIV and trained in IBSR to conduct the training of trainers.

Nine studies described having regular supervision meetings weekly (Balaji et al. 2011; Mathias et al. 2019; Dow et al. 2020; Osborn et al. 2021; Simms et al. 2022; Venturo-Conerly et al. 2022b, 2024; Osborn et al. 2023) and monthly (Tinago et al. 2024) to review previous sessions as well as prepare for upcoming sessions. Nine studies explicitly stated the availability of adult coaches or research staff at each intervention session to provide support where needed (Balaji et al. 2011; Mathias et al. 2018; Osborn et al. 2021; Venturo-Conerly et al. 2022b, 2024; Ferris France et al. 2023; Harrison et al. 2023; Merrill et al. 2023; Osborn et al. 2023). This includes youth leaders facilitating alongside community health counsellors (Im et al. 2018), team leaders stepping in if a youth leader was absent (Mathias et al. 2018) or unable to deal with a situation outside their scope of knowledge (Merrill et al. 2023), and readily available care in case of potential risk identified during the intervention (Venturo-Conerly et al. 2024). Two studies regularly utilised fidelity checklists throughout the study to ensure intervention quality (Dow et al. 2020; Ferris France et al. 2023). Three studies

**Table 2.** Description of peer leaders

| Author and year | Characteristics of peer leaders | Number of peer leaders | Recruitment process, training, supervision, support, remuneration, safe-guarding | Responsibilities and tasks |
|---|---|---|---|---|
| **Psychotherapy and counselling** | | | | |
| Dow et al. 2020 | Young adults (23–30 years old) with background of either living with HIV and/ or having prior experience with mental health research. | N = 6 | Recruitment: Not specified<br>Training: On-site intensive two-week training with the principal investigator and a U.S. based clinical psychologist.<br>Supervision, support, safeguarding: Group leaders continued to have weekly practice under supervision 1 day before the intervention group sessions. This is followed by a review of session notes and fidelity checklists.[*1]<br>Remuneration: Not specified | Group leaders were sex-concordant with participants in their group. Two group leaders led each session, and the third group leader would be responsible for keeping detailed notes of each youth and ensured protocol delivery using a fidelity checklist. |
| Ferris France et al. 2023 | Community Adolescent Treatment Supporters (CATS) are 18–24 year olds LHIV. | N = 15 | Recruitment:15 CATS were identified out of a group of 30 CATS initially recruited to receive the intervention delivered by adult coaches. Selection process not specified.<br>Training: A residential face-to-face, 6-day immersive Training of Trainers, led by local adult coaches LHIV trained in IBSR.<br>Supervision, support, safeguarding: CATS were supported by the local adult coaches at each intervention session. Each day that training and/or intervention were evaluated, feedback was used real-time to adapt the curriculum. Investigators assessed and graded group leader competence and intervention fidelity<br>Remuneration: Not specified | CATS peer coaches delivered the second round of the intervention to 30 CATS participants. |
| Osborn et al. 2020a | Group leaders 17–21 years old who are high school graduates and required to read in English. None were formally certified in counselling or related fields. | N = 5 | Recruitment: Group leaders were recruited from local universities and high-school graduate forums with a written application and interview process assessing past experiences, interest in the project, familiarity with mental health issues, and interpersonal facilitation skills. Five out of 11 applicants were selected.<br>Training: 20 h of training spanning 5 days and covering the content of both Shamiri and study skills group. Training was led by the first three authors involving general communication and leadership skills, handling conflicts and referring students to appropriate school officials. This is followed by didactic training in the specific content of the weekly intervention sessions. The group leaders were then asked to role-play delivering the intervention content, and they received feedback from trainers and fellow group leaders.<br>Supervision, support, safeguarding: Not specified<br>Remuneration: Not specified | Group leaders were instructed to strictly follow the protocol manual and not use content from the other condition during sessions. All group leaders led their groups individually. Three were assigned to the Shamiri intervention, two to the study skills control. |
| Osborn et al. 2021 | Group leaders were high school graduates fluent in English and Kiswahili 18–26 years old. | N = 13 (61.54% female) | Recruitment: Selection via a semistructured interview that gauged past experiences and interpersonal skills<br>Training: 10 h of training by the study team covering: Shamiri and study skills intervention content, counselling techniques, role play, control content, and safety protocols.<br>Supervision, support, safe-guarding: Group leaders had weekly supervision phone calls with study staff to ask questions, receive feedback, and bring up any concerns. Study staff are present during data collection and intervention delivery for support. In the event of serious mental health concern, group leaders are trained to call their supervisor.[*2]<br>Remuneration: Not specified | All leaders led both Shamiri and study skills content |

*(Continued)*

**Table 2.** (*Continued*)

| Author and year | Characteristics of peer leaders | Number of peer leaders | Recruitment process, training, supervision, support, remuneration, safe-guarding | Responsibilities and tasks |
|---|---|---|---|---|
| Osborn et al., 2023 | Youth facilitators 18–22 years old from the Nairobi area and have a high school diploma. | Not specified | Recruitment: Semi-structured interviews were conducted, and past experiences, interests, interpersonal skills were assessed, following a validated protocol for youth lay-providers widely used in Kenya (Venturo-Conerly et al. 2022a)<br>Training: They received 20 h of training covering the Pre-Texts and study skills content, led by the first author. Topics covered include general communication and group facilitation, referring students in need to proper school resources. The recruitment and training followed validated protocol for youth lay providers widely used in Kenya.<br>Supervision, support, safe-guarding: Weekly supervision meetings were provided to review sessions that would take place in the coming week, and address any concerns faced in the past week. A member of the study team was assigned as support to provide materials, give time warnings and available in the general area if any concerns should arise. WhatsApp was used for any important reminders.[*3]<br>Remuneration: A stipend of $150, and full reimbursement for transportation.[*4] | Facilitators were randomly assigned to groups in each school, with each facilitator leading both Pre-Texts and study skills groups. They were tasked with referring students in need to proper school resources. They also facilitated the use of challenging texts as raw material for artmaking. |
| Simms et al. 2022 | Community Adolescent Treatment Supporters (CATS) who are 18–24 years old living with HIV who are trained and mentored to provide peer counselling and support. | N = 60[*5] | Recruitment: CATS are selected based on readiness and capacity to provide peer counselling<br>Training: A three-week programme utilising role play, pre- and post-tests, group sessions and one-on-one sessions for those with difficulty. They are trained on the use of PST with ALHIV and how to make referrals to mental health services.[*5]<br>Supervision, support, safe-guarding: There was weekly nurse-led group supervision and monthly supervision from a mental health specialist.[*5] CATS in Zvandiri-PST arm met a Zvandiri mentor at least once every 2 weeks to review individual cases<br>Remuneration: Not specified | Tasks include helping participants choose a manageable, relevant problem, establish a goal, and brainstorm solutions. Afterwards, selecting a detailed solution and devising an action plan. |
| Venturo-Conerly et al., 2022b | Lay-providers that were recent Kenyan high school graduates. | N = 13 | Recruitment: Lay-providers were selected based on online applications and in-person interviews. They were assessed for interest, availability, conscientiousness, experience, personal characteristics, and experience indicating good leadership abilities.<br>Training: Lay-providers were trained as part of Osborn et al., 2020a study (see above) in the four-session format of the Shamiri intervention, but also an additional hour-long training in how to deliver the single-session versions for this study. All lay-providers were trained in all conditions to allow for random assignment to conditions<br>Supervision, support, safe-guarding: Weekly supervision meetings were provided to review sessions that would take place in the coming week, and address any concerns faced in the past week. A member of the study team was assigned as support to provide materials, give time warnings and available in the general area if any concerns should arise. WhatsApp was used for any important reminders.[*6]<br>Remuneration: A stipend of $150, and full reimbursement for transportation.[*6] | Different lay-providers were assigned to facilitate three intervention components: Growth Intervention, Gratitude Intervention, Value Affirmation Intervention, and Study-Skills Control. In the Growth Intervention, lay-providers explained that everyone can improve with effort, followed by participants engaging in planned activities. Finally, lay-providers presented a take-home activity where participants used skills and concepts from the session to address a personal challenge.<br>In the Gratitude Intervention, lay-providers began the session by emphasising the importance of gratitude and discussing personal examples. Participants then engaged in planned activities. Finally, lay-providers assigned a take-home activity: writing three things for which participants felt grateful every day for 1 week.<br>In the Value Affirmation Intervention, lay-providers |

**Table 2.** (*Continued*)

| Author and year | Characteristics of peer leaders | Number of peer leaders | Recruitment process, training, supervision, support, remuneration, safe-guarding | Responsibilities and tasks |
|---|---|---|---|---|
| | | | | described how personal values contribute to shaping our lives, including decision-making and academic outcomes. Participants then engaged in planned activities. Finally, lay-providers assigned a take-home activity involving a specific values-promoting exercise planned previously. |
| Venturo-Conerly et al., 2024 | Lay-providers that were recent Kenyan high school graduates aged 18–21. | *N* = 20 | Recruitment: Lay-providers were selected using semi-structured interview assessing experiences with youth, interpersonal skills, and attitudes toward mental health. Training: Four days of teachings on protocols, general counselling techniques, ethical guidelines for research, and risk procedures. Training included clinical risk assessment and response, sensitive to local resources, attitudes and customs, and were involved in the monitoring of adverse events that may require escalation to a higher-level doctoral psychologist. Supervision, support, safe-guarding: Each lay-provider was assigned a clinical supervisor with at least a bachelors-level of psychology and counselling experience, who trained them, supervised group sessions, and provided care in case of risk. Weekly supervision and daily onsite supervision were implemented. Remuneration: A stipend of $150, and full reimbursement for transportation.*[7] | Each trained lay-provider facilitated a group of approximately 8–15 students. Lay-providers facilitated three interventions: (1) the Growth Intervention, which challenges the belief that personal characteristics are fixed, teaching participants that growth and improvement are possible, especially when facing challenges; (2) the Gratitude Intervention, designed to enhance feelings and expressions of gratitude to increase well-being and improve relationships; and (3) the Values Intervention, which encourages students to cultivate their personal values and intentionally plan and act in alignment with these values. |
| **Peer education and psychoeducation** | | | | |
| Balaji et al. 2011 | Not specified | Rural communities: *N* = 28 trained, 20 attended more than 75%. Educational institution-based components: *N* = 4 even though 98 peer leaders trained, 75 attended more than 75% | Recruitment: Selected based on a pre-determine non-specified criteria Training: Training was led by psychologists and social workers experienced in adolescent health. Training materials were from standardised manuals. Resources guide was given for delivering the intervention. Supervision, support, safe-guarding: Support was provided in the form of a Community Advisory Board consisting of village council leaders and trained teachers. There were on-site supervision and weekly review meetings. Remuneration: moderate monetary and other incentives (certificates) | Peer leaders were given a resource guide for delivering the intervention. They conducted group sessions and performed street plays. |
| Im et al. 2018 | Community youth leaders in the Somali community. | *N* = 10 (initially 25 trained) | Recruitment: Not specified Training: A weeklong TIPE training of trainer (TOT) by the project team, followed by training on facilitation and program monitoring and evaluation. Supervision, support, safe-guarding:: Not specified Remuneration: Not specified | 10 trained youth leaders were paired with five community health counsellors to perform the intervention. |
| Kermode et al. 2021 | Peer facilitators were individuals affected by mental illness, with at least Class 12 education, ability to travel to intervention sites and communication skills. | *N* = 8 | Recruitment: From local communities (Details not specified) Training: Training was led by the Burans Community Mental Health Project Team. (Details not specified) Supervision, support, safe-guarding: Support was offered by the Burans team. (Details not specified) Remuneration: Not specified | Two peer facilitators led each group (11 groups of ~13 participants) |

**Table 2.**  (*Continued*)

| Author and year | Characteristics of peer leaders | Number of peer leaders | Recruitment process, training, supervision, support, remuneration, safe-guarding | Responsibilities and tasks |
|---|---|---|---|---|
| Mathias et al. 2018 | Group facilitators were locally recruited young women 20–30 years old expressing enthusiasm to work in youth resilience and have completed 12<sup>th</sup> Class. | Not specified | Recruitment: Not specified<br>Training: 5 days of training and 2 days of refresher training after the completion of eight modules.<br>Supervision, support, safe-guarding: They were supported by a research team leader on tasks including planning, reporting, and who stepped in if a peer facilitator was unable to attend. The Nae Disha field coordinator supported all aspects of the intervention delivery and performed monitoring and support.<br>Remuneration: Not specified | Facilitators led in pairs with eight groups of 12–15 participants for 15 consecutive weeks. |
| Mathias et al., 2019 | Peer facilitators were selected from the four target communities, who were young people aged under 30 years, with personal experience of mental ill-health and who had completed 12<sup>th</sup> class in high school. | N = 8 | Recruitment: Facilitators were selected through a process of community meetings and interviews with community leaders.<br>Training: Peer facilitators were trained for five days in group facilitation, how to support young people with PSD, and in the Nae Disha curriculum. Refresher training was conducted across the five-month intervention period.<br>Supervision, support, safe-guarding: There was ongoing supervision by project staff.<br>Remuneration: Peer facilitators were paid. (Amount not specified) | Facilitators conducted 17 interactive sessions. It included modules such as accepting differences, managing emotions, communicating confidently, protecting ourselves etc. Facilitators also encouraged participants to participate in collective activities, direct participants to additional mental health services, and visit a de-addiction centre. |
| Merrill et al., 2023 | Youth Peer Mentors (YPM), aged 21–26 years old selected by healthcare providers as successfully managing their HIV were hired to work in the clinics. | Not specified | Recruitment: Process not specified.<br>Training: Completed an intensive two-week training and underwent 1 month of practice meetings with youth before the intervention launch.<br>Supervision, support, safe-guarding: Healthcare providers (HCPs) led the orientation and caregiver meetings and were available to answer questions if outside the scope of the YPM's knowledge.<br>Remuneration: YPM were paid for their work. (Amount not specified) | YPM were introduced to their youth in the orientation meeting where they discuss action plans for subsequent meetings again. They have monthly individual and group meetings with youth following the Project YES! curriculum. |
| Mohamadi et al. 2021 | Peer educators were active volunteers who scored higher on the puberty health questionnaire prior to the start of the study. | Not specified | Recruitment: Based on scores on the puberty health questionnaire.<br>Training: Educational content was taught to peer educators in one session.<br>Supervision, support, safe-guarding: Peer-to-peer educators' relationship with researcher continued so that educators could ask questions.<br>Remuneration: Not specified. | Each peer educator was responsible for transmitting information to 5–6 other students in one formal session, followed by informal sessions in small groups within 1 month. |
| Yuksel et al. 2019 | Mentors were fourth year nursing students with high academic achievement and effective communication skills, be loved among friends and volunteered to participate in the studies. | N = 10 | Recruitment: Among students who volunteered, those with an academic grade point of 2.50 or higher were ranked from high to low. Faculty members were then interviewed to determine the students who had effective communication skills and who were loved by their friends.<br>Training: Mentors received 10 h of training over five days, including the features of peer counselling, communication, assistive communication techniques, and coping with stress.<br>Supervision, support, safe-guarding: Researchers continued to provide guidance to the mentors throughout the eight weeks.<br>Remuneration: Not specified | Two mentors were assigned to each group of eight to 10 mentees. The mentors applied eight weekly sessions of peer mentoring to the mentees. |

**Table 2.** (*Continued*)

| Author and year | Characteristics of peer leaders | Number of peer leaders | Recruitment process, training, supervision, support, remuneration, safe-guarding | Responsibilities and tasks |
|---|---|---|---|---|
| **Peer support** | | | | |
| Duby et al. 2021 | Trained peer-educators of similar age | Not specified | Not specified | Not specified |
| Harrison et al., 2023 | Peer mentors who also live with a chronic condition | Not specified | Recruitment: Process not specified. Training: Peer mentors are trained to adjust their topics to age and cognitive maturity of the group. Supervision, support, safe-guarding: Peer mentors are overseen by a supervisor (either a psychologist or social worker) at each session to provide additional support. Remuneration: Not specified. | Volunteer peer mentors are tasked with approaching patients in the Groote Schuur Hospital (GSH) waiting room and inviting them to participate in the Better Together group sessions. These group sessions are designed to help adolescents with chronic conditions (including HIV, renal disease and psychiatric conditions) to (1) build social networks that enhance psychosocial support; (2) develop a sense of belonging with peers; (3) create a space where adolescents can share their experience(s) living with and managing chronic illness; and (4) build empathy among Adolescents Living with HIV (ALHIV) and other conditions. |
| Tinago et al., 2023 | Peer educators are women who had given birth during adolescence, between the ages 19 and 25, with at least seventh grade education. | $N = 12$ | Recruitment: Peer educators were recruited by project coordinator and community health workers through snowball sampling and in-person recruitment. Training: They were trained by the project coordinator and local subject matter experts with a 3-day training session. Supervision, support, safe-guarding: WhatsApp was used as additional support for training and implementation and a platform to communicate and plan sessions. Monthly meetings were conducted to review session plans and project progress. Remuneration: Not specified. | Along with Community Health Workers (CHWs), peer educators co-facilitated peer support groups which discussed 12 participant identified topics: (1) introduction to the peer support groups, (2) adolescent motherhood, (3) gossip, (4) healthy relationships, (5) depression, (6) substance abuse, (7) family planning, (8) sexual health, (9) healthy parenting, (10) income generation, (11) hygiene, and (12) moving forward as an adolescent mother. |

*Information extracted from protocol paper:

[1] Dow DE, Mmbaga BT, Turner EL, Gallis JA, Tabb ZJ, Cunningham CK, et al. Building resilience: a mental health intervention for Tanzanian youth living with HIV. *AIDS Care.* 2018;30(sup4):12–20.

[2] T.L. Osborn, K.E. Venturo-Conerly, G.S. Arango, E. Roe, M. Rodriguez, R.G. Alemu, J. Gan, A.R. Wasil, B.H. Otieno, T. Rusch, D.M. Ndetei, C. Wasanga, J.L. Schleider, and J.R. Weisz, Effect of Shamiri Layperson-provided intervention vs. study skills control intervention for depression and anxiety symptoms in adolescents in Kenya: A randomised clinical trial. *JAMA Psychiatry* 78 (2021) 829–837. (Supplementary Materials).

[3] T.L. Osborn, D.M. Ndetei, P.L. Sacco, V. Mutiso, and D. Sommer, An arts-literacy intervention for adolescent depression and anxiety symptoms: outcomes of a randomised controlled trial of Pre-Texts with Kenyan adolescents. *EClinicalMedicine* 66 (2023) 102288.

[4] K. Venturo-Conerly, E. Roe, A. Wasil, T. Osborn, D. Ndetei, C. Musyimi, V. Mutiso, C. Wasanga, and J.R. Weisz, Training and supervising lay providers in Kenya: Strategies and mixed-methods outcomes. *Cognitive and Behavioural Practice* 29 (2022a) 666–681.

[5] S. Chinoda, A. Mutsinze, V. Simms, R. Beji-Chauke, R. Verhey, J. Robinson, T. Barker, O. Mugurungi, T. Apollo, and E. Munetsi, Effectiveness of a peer-led adolescent mental health intervention on HIV virological suppression and mental health in Zimbabwe: protocol of a cluster-randomised trial. *Global Mental Health* 7 (2020) e23.

[6] K.E. Venturo-Conerly, T.L. Osborn, R. Alemu, E. Roe, M. Rodriguez, J. Gan, S. Arango, A. Wasil, C. Wasanga, and J.R. Weisz, Single-session interventions for adolescent anxiety and depression symptoms in Kenya: A cluster-randomised controlled trial. *Behaviour Research and Therapy* 151 (2022b) 104040.

[7] K.E. Venturo-Conerly, A.R. Wasil, T.L. Osborn, E.S. Puffer, J.R. Weisz, and C.M. Wasanga, Designing culturally and contextually sensitive protocols for suicide risk in global mental health: Lessons from research with adolescents in Kenya. *J Am Acad Child Adolesc Psychiatry* 61 (2022c) 1074–1,077.

(Osborn et al. 2023; Tinago et al. 2024; Venturo-Conerly et al. 2022b) used WhatsApp, an online instant messaging application, as an adjunct to support communications, while Mohamadi et al. (2021) and Yuksel and Bahadir-Yilmaz (2019) mentioned an ongoing relationship between peer leaders and the research team.

Only six studies (Balaji et al. 2011; Mathias et al. 2019; Venturo-Conerly et al. 2022b, 2024; Merrill et al. 2023; Osborn et al. 2023) detailed a form of remuneration for the peer leaders. Osborn et al. (2023) and Venturo-Conerly et al. (2022b, 2024) paid a stipend of $150 and covered transportation costs throughout the intervention period (Venturo-Conerly et al. 2022b), Balaji et al. (2011) provided "moderate monetary" and other incentives (i.e. certificates), while Mathias et al. (2019) and Merrill et al. (2023) simply stated that peer leaders were paid (without including the amount).

### Key findings in mental health outcomes

The studies used various instruments to measure mental health outcomes. General mental health symptoms were measured using the General Health Questionnaire-12 (GHQ-12), Strengths and

Difficulties Questionnaire (SDQ), Symptom Checklist-25 (SCL-25), Social Support Questionnaire (SSQ), WHO-5 Well-being Index, Psychological Outcomes Profile, Beck Youth Inventories Second Edition (BYI-II), Child Attitudes Toward Illness Scale (CATIS), Short Warwick-Edinburgh Mental Wellbeing Scale (SWEMWBS), Shona Symptom Questionnaire (SSQ), Persian Standard Symptom Checklist-25 (SCL-25). Anxiety was measured with the Generalised Anxiety Disorder-7 (GAD-7), depression with the Patient Health Questionnaire -9 and -8 (PHQ-9, PHQ-8), and post-traumatic stress using the UCLA PTSD Reaction Index survey, PTSD Check List – Civilian Version (PCL-C). Other mental health scales include the Connor-Davidson Resilience Scale (CD-RISC), Schwarzer general self-efficacy scale, Multidimensional Scale of Perceived Social Support (MSPSS), Perceived Control Scale for Children (PCS), Ways of Coping Inventory (WCI), and Peer and Significant Adult Support (PSAS).

All quantitative studies reported that peer-led interventions led to improvements in at least one mental health symptom scale (Table 1). Studies exploring general mental health symptoms reported an improvement in scores for mental health symptoms, probable depression and suicidal behaviour, as well as an intervention effect on the prevalence and severity of common mental health outcomes. Studies measuring depression and anxiety reported a larger decrease in scores in the intervention than the standard of care or active control groups. Two studies examined PTSD symptoms and reported an improvement in scores in the intervention group (Im et al. 2018; Dow et al. 2020). None had reported worsening of mental health as a result of the intervention.

Qualitative studies similarly reported improvements in mental health (Table 3). There were reports of improved self-esteem, confidence and self-worth (Duby et al. 2021; Ferris France et al. 2023; Merrill et al. 2023), combating feelings of isolation with community building (Mathias et al. 2019; Harrison et al. 2023; Merrill et al. 2023), and improved mental health (Mathias et al. 2019).

### *Cultural considerations during implementation*

Several studies specified consideration of the respective country's sociocultural context during development of their intervention. (Im et al. 2018; Mathias et al. 2018; Dow et al. 2020; Osborn et al. 2020a; Osborn et al. 2021). This entailed, for example, conducting group sessions with YPLHIV in same-sex groups with sex-concordant peer leaders (Dow et al. 2020) or adopting common colloquial terms related to mental health (cultural idioms) identified by local community partners to avoid pathologising trauma responses with western terminology (Im et al. 2018). However, several studies also acknowledged the use of mental health measures that have not been validated or adapted to the cultural context was inevitable due to the lack of alternatives (Im et al. 2018; Dow et al. 2020; Osborn et al. 2020a; Osborn et al. 2021).

### *The mechanisms that make peer-led interventions effective and their implementation challenges*

Several studies reported specific aspects of having a peer-led component, detailing mechanisms that made the intervention more effective than if it had been delivered by adults or non-peers (Table 4). Such mechanisms include the provision of a unique form of emotional support, where youth could relate on a peer-to-peer level (Duby et al. 2021; Simms et al. 2022). Participants similarly appreciated the ability to connect with peer leaders given their similar age and lived experiences, for instance, the shared

experiences of living with HIV (Merrill et al. 2023). There was also a preference for peer delivery when it came to sensitive topics such as Sexual and Reproductive Health (SRH) (Duby et al. 2021; Merrill et al. 2023; Tinago et al. 2024). The enhanced sense of community was well received, with participants describing the alleviation of social isolation and increased social support (Im et al. 2018; Harrison et al. 2023; Tinago et al. 2024). In some studies, participants praised positive qualities of peer leaders, such as patience, kindness and respect (Mathias et al. 2019). Studies also reported the added benefit of peer-led interventions to the peer leaders themselves, that they increased their self-confidence and leadership ability (Balaji et al. 2011; Ferris France et al. 2023), and allowed them to learn from the participants and gain new experiences (Harrison et al. 2023).

However, several challenges to peer-led interventions were also reported. There were issues with adherence of peer leaders to the intervention, with many citing other commitments to school or housework, inconvenient locations and timings (Balaji et al. 2011). Furthermore, integrating a peer-led system into existing structures is expensive and requires much support from institutions to overcome logistical barriers (Osborn et al. 2021; Ferris France et al. 2023). This was similarly echoed by Kermode et al. (2021), who reported that a lot of support was required by the project team to ensure the success of the peer-led intervention. Harrison et al. (2023) described an emotional toll that peer leaders face when sharing their own stories and striving to understand and support participants, further underscoring the importance of providing adequate support to peer leaders.

### Discussion

There is a critical need to address the youth mental health gap in LMICs, and task-sharing with peer leaders present a possible solution. The purpose of this review was to identify intervention trials that utilised peer-led approaches to address youth mental health in LMICs, and to synthesise the types of interventions, delivery components, effectiveness, benefits and challenges encountered during implementation. The inclusion of pilot studies and quasi-experimental trials contributed to an overview of peer-led mental health interventions that are currently in the developmental phase. Our review suggests that a diverse range of peer-led interventions can generate positive improvements in mental health among adolescents in LMICs.

### *Peer-led interventions can improve mental health outcomes of youth living in LMICs*

Peer-led interventions included in this review were highly heterogeneous in terms of design, mode of delivery and content. Yet, all reported improvements in different mental health outcomes, highlighting their versatility to be adopted in many contexts. Within peer psychotherapy and counselling interventions, studies used either group and individual therapy or didactic lectures and non-didactic methods (i.e. music reflection, poem, group discussions, podcast making) to increase youth engagement (Balaji et al. 2011; Osborn et al. 2020a; Duby et al. 2021; Osborn et al. 2021). Previous research has demonstrated that peer-delivered psychotherapy is effective for youth living in LMICs (Singla et al. 2021; Tomfohr-Madsen et al. 2022) and outperforms waitlist and active control conditions (Venturo-Conerly et al. 2023). For vulnerable individuals facing stigmatising mental illnesses, peer delivery may be the most acceptable and most culturally appropriate method to receive evidence-based treatment (Tomfohr-Madsen et al. 2022). Peer education sessions adopted group classroom sessions, street plays, participation in community

**Table 3.** Mental health outcomes and quotes from qualitative papers

| Author and year | Qualitative outcomes | Quotes |
|---|---|---|
| Duby et al. 2021 | • Building self-esteem, social confidence<br>• Empowerment, self-worth and self-respect<br>• Improved well-being and coping through communicating emotions | • "The clubs helped my self-esteem… (participating in the clubs) really helped, a lot… for me to feel that my self-esteem is better." AGYW 15–18 years<br>• "The club helped me gain confidence" AGYW 15–18 years<br>• "The Rise Club awareness has taught use to respect ourselves as women… you must respect your own body to show others that they must respect you as a woman.<br>• "I was taught to open up, and now I can speak up for myself and I stop keeping quiet"<br>• I used to say I prefer to stay alone with my problem, and it stresses me… but in these programmes, sharing, talking… it helped me… to share my story. |
| Ferris France et al. 2023 | • Self-confidence and self-agency<br>• Sense of purpose and meaning in life<br>• Body positivity<br>• Improved communication and personal/family relationships<br>• Self-forgiveness and forgiveness of others | • "After the training I learnt that there is nothing I cannot do, and I can do it" Male 21 years, FGD<br>• "It was only after Wakakosha when I realised that I should be proud of my body" Female 23 years, IDI |
| Harrison et al., 2023 | • An eye-opening and powerful experience, combating feelings of isolation<br>• Combating stigma and uninformed attitudes, finding support and acceptance in the support group | • "I mean, it is nice to have a group of people with the same illness, but to me I feel like it is nice to get to chat with people with different illnesses because you hear different experiences from them and even though we may think that it is different, but we have many similarities and things that we go through." (PID 00001, Male age 20, living with renal disease)<br>• "I live with this thing that cannot be cured. …But I learned to accept it because I thought to myself, you know what? I can look at this from a different point of view. I thought about so many people who are out there successful, who are HIV positive. Because it does not mean a death sentence when you are HIV positive. (PID 00015; Female Age 22, living with HIV." |
| Mathias et al., 2019 | • Formation of new peer friendship networks<br>• Increased self-efficacy<br>• Improved mental health<br>• Greater confidence in communicating | • "At the group we played with balloons and likewise, we now play balloon games with the children at our home. When three or four of us got together we laughed and then we felt happy."<br>• "There was one girl who used to be very quiet and not speak but through coming to this group she now speaks confidently…This change happened in her because she could share easily with us about her problems, and because of playing games and doing role plays." |
| Merrill et al., 2023 | • Overcoming shame and developing greater feelings of self-worth<br>• Community building | • "Sometimes even on our own you have a certain stigma. You feel embarrassed. You don't feel free in your own life. But for me, [the program] has taught me a lot. I can even stand in public and talk about my status with confidence." Female 24 years old<br>• "Project YES! has actually opened my eyes to see we are not alone. I am not alone." Male 21 years |

activities and one-to-one meetings. Peer education has been reported to be enjoyable and preferred, and can even be more effective than information transmission by professionals, especially when it comes to sensitive topics (Topping 2022). This review also suggests that the provision of a peer-support network was enough to alleviate some mental health symptoms. Social support serves as a way to build an in-group community to mitigate social isolation, which is associated with poor mental health outcomes (Grønlie and Dageid 2017; Leigh-Hunt et al. 2017). The perception of social support is sometimes more important than the actual support received, and acts as a coping mechanism for daily stressors and a protective mechanism against the development of more severe psychological distress (Casale et al. 2019). The key benefit of peer support is the mutual benefit to all parties involved in giving and receiving support (Arndt and Naudé 2020).

### Key components to cultivate effective peer-leaders

The TRUST framework (Training, Referral pathways, Understanding the remit of their role, Supervision, and recognition that Talking helps) outlines the needs of young peer supporters (Wogrin et al. 2021; Simms et al. 2022). Training of peer leaders was mentioned in all studies included in this review, and was often conducted directly by the research team or by adult coaches (Ferris France et al. 2023). While training is critical for adult and youth leaders alike, it was especially important for youth leaders as they were often younger and had less education and work experience compared to adults (Maticka-Tyndale and Barnett 2010). To mitigate this, it was helpful to provide guidance on the topics of health literacy, cultural practices, mental health, as well as the soft-skills of communication, leadership and conflict resolution. Refresher trainings, included in two studies (Mathias et al. 2018; Mathias et al. 2019), have been shown to be useful for longer interventions to ensure reinforcement of accurate information, adjustments and feedback (Maticka-Tyndale and Barnett 2010). Similarly, many studies involved regular supervision sessions to ensure proper conduct of intervention sessions with an opportunity to adapt intervention delivery based on feedback. Effective supervision was critical, and studies without supervision showed diminished quality and fidelity in intervention delivery (Sharma 2002). Youth engagement can be challenging and emotionally draining (Simms et al. 2022), and regular supervision with the research team can help prevent the development of secondary trauma and ensure the welfare of the youth leaders. Osborn et al. (2023) provided a relevant protocol for the training and supervision of lay youth providers, which can guide future research and implementation of peer-led interventions.

**Table 4.** The mechanisms that make peer-led interventions effective and their implementation challenges

| Author and year | Benefits | Challenges |
|---|---|---|
| Balaji et al. 2011 | Peer leaders report increase in self-confidence and leadership ability. There was also report of greater anger control, communication skills, reduced smoking and greater comfort in discussing sexual health issues. | Non-adherence of peer leaders due to school, house-work, other commitments, inconvenient timings, duration and locations. Integration of peer education into existing structures to ensure long term sustainability – peer education is expensive, logistical barriers. |
| Duby et al. 2021 | (1) The provision of emotional support and counselling<br>(2) Comfort and ability to relate on a peer-to-peer level<br>(3) Preference for receiving SRH from a similar age peer rather than an older adult | |
| Ferris France et al. 2023 | CATS felt a transformative effect of delivering the intervention themselves and learnt new skills | Covid–19 and delivering intervention online. Lack of face-to-face context and poor network connection, erratic power supply, exorbitant data charges. |
| Harrison et al., 2023 | Vital importance of social support provided by peer mentor and peer support program<br>"It teaches me a lot about other illnesses and how to approach or interact with others"<br>Peer mentors themselves get to learn from the participants and new experiences. | An emotional toll when interacting with other young people with chronic illness, including sharing one's story and working to understand and provide support to others |
| Im et al. 2018 | TIPE intervention helped build a support system and enhanced the sense of community among participants. It promoted perceived social support, particularly among those with high PTSD symptoms. | |
| Kermode et al. 2021 | | The peer facilitators required a lot of support from the project team, which was essential for the success of the intervention |
| Mathias et al., 2019 | Young people were willing to participate and continue the intervention as they were assured of respect and kindness from peer facilitators "You never belittled our talk but always built it up. We came because we understood what you said, otherwise we wouldn't have come. Who cares for drug addicts these days? But you showed that you did." | |
| Merrill et al., 2023 | Individual meetings with youth peer mentor (YPM) gave opportunity to open up about personal issues or sensitive topics like condom use and sexual behaviour. YPM unique abilities to connect with youth given a similar age and shared experience of living with HIV | |
| Simms et al. 2022 | Peer-led intervention created a rare forum for adolescents to legitimately discuss their problems and be listened to with empathy by a trusted peer | Limits to how youth can solve their problems themselves as they have limited influence over their situations if they do not have adult support. |
| Tinago et al., 2024 | Peer groups provided new forum to address primary sources of stigma, sharing challenges and seeking advice, alleviating social isolation as a source of stress<br>Sole source of sexual and reproductive health information<br>Consistent community engagement and trust-building efforts lead to increasingly positive and supportive attitudes toward participants and the intervention concept | |

## Cultural appropriateness and generalisability in reporting mental health

Cultural appropriateness allows for the acceptability of an intervention, affecting the effectiveness and the generalisability of study outcomes. There is a lack of culturally validated mental health scales in LMICs, and many studies currently use scales validated in HICs (Kaiser et al. 2022). This makes it difficult to effectively assess and generalise quantitative clinical improvement in mental health in LMICs. Qualitative methods of assessing mental health interventions may be beneficial in such circumstances, and Ferris France et al. (2023) had justified their intentional use of a qualitative rather than quantitative assessment of their intervention by likening the comparison of health outcomes scales across different cultures to compare chopsticks with forks. Moreover, mental health is still a highly stigmatised topic, which can affect participant reporting of symptoms and therefore study outcomes. Several studies reasoned a low baseline report of symptoms due to

reluctance in reporting mental health difficulties (Im et al. 2018) and challenges in articulating feelings and emotions (Ferris France et al. 2023). Venturo-Conerly et al. (2022c) has adapted principles from the Belmont Report (Department of Health et al. 2014) and Declaration of Helsinki (World Medical Association 2013) to form guidelines for addressing risk in high-stigma environments. Methods that can ensure cultural appropriateness would be including local mental health providers familiar with local resources, regulations and norms, who can identify nuances of distress that may indicate risk. Training lay providers in risk assessment and the appropriate channels for escalation and management is also crucial.

## Sustainability of peer-led interventions

Ideally, once an intervention is implemented and effective, it would continue without reliance on the research team. For this to happen,

it needs to be integrated within existing structures through partnerships with governments and established institutions. This is the goal of many interventions (Im et al. 2018). Some studies reported that benefits of the intervention might revert back to baseline once the group stops meeting regularly (Mathias et al. 2018; Kermode et al. 2021). However, facilitating such institutional or governmental partnerships is often a challenge (Kelly et al. 2006; Maticka-Tyndale and Barnett 2010). This is echoed by several studies that listed integrating peer education into existing educational institutions as a barrier, citing the lack of support for peer educators and the research team, and poor cooperation from teachers and school administrators (Balaji et al. 2011; Mohamadi et al. 2021). Osborn et al. (2021) reported an unexpected government ban on research activities in educational institutions that limited the assessment of an extended time point and led to significant attrition in participant attendance. Several studies (Dow et al. 2020; Osborn et al. 2020a; Osborn et al. 2021) have integrated peer-led interventions into stepped-care models, where at-risk youth are referred to trained adult clinical providers within established care systems to address elevated symptoms and emergencies. Peer-led models do not function in isolation; their integration into existing care systems ensures a more seamless process for connecting new interventions with established frameworks. However, the reliance of sustainability on external organisations leaves them vulnerable to sudden policy changes and motivations of the existing structures. It is therefore vital for research teams to empower the youth and their communities and establish delivery mechanisms that reduce barriers to these partnerships. To overcome such challenges, Dow et al. (2020) developed a protocolised manual allowing for scalability and reproducibility, while Ferris France et al. (2023) provided an online Toolkit to offer continuous support and mentoring to peer coaches.

### Benefits and challenges to task shifting to peer-leaders

The benefits and challenges of a peer-led intervention reported by studies in this review has similarly been described in the literature. Peer-led interventions allow for the contribution of lived experience, which makes interventions more equitable and contextually acceptable. Peers have been used globally in the delivery of maternal mental health care, and a systematic review by Atif (2015) reported that peers who were mothers themselves were perceived as more trustworthy, friendly and experienced and likely to share personal insights to benefit other mothers (Nankunda et al. 2010). Moreover, the study reported that peer openness about their health status contributed to overcoming illness-related stigma (i.e. HIV status). Peer-led interventions are beneficial to adolescents as they provide a unique form of rapport that allows discussions of sensitive topics such as sexual health (Visser 2007). Adolescents frequently relate more easily with a peer and communicate in a language that is understandable and accessible (Visser 2007). Positive peer interactions also allow for role modelling, and often peer leaders are viewed as educators in their communities who can impart contextually relevant information (Atif 2015). A key benefit of a peer-led component is the reciprocity and mutual benefit to the giver and receiver of the intervention (Naudé 2017). Positive peer interactions are associated with higher self-esteem, wellbeing and health literacy for all parties involved in the delivery of a peer-led intervention (Naudé 2017). The act of delivering an intervention can also improve confidence, fulfilment and life skills, and can even

improve an individual's social status and mobility within communities (Alcock et al. 2009; Atif 2015).

Barriers to implementing a peer-led interventions include ensuring a stable workforce. Peer leaders may leave for university, seek other employment opportunities or eventually age out of the role, requiring ongoing training of an evolving workforce (Okoroafor and Dela Christmals 2023). In addition, adolescents often lack full control over their time and environment, with non-adherence commonly resulting from school, home commitments and travel. Balaji et al. (2011) emphasises the importance of incentives such as remuneration, certificates, and prizes to help support the project. Providing compensation also allows lay providers to devote time and energy to intervention training and delivery, freeing them from the need for alternative employment (Venturo-Conerly et al. 2022a). Peer facilitators require significant support from the team, including time for training and supervision, as well as mentorship from trained staff (Kermode et al. 2021). Although lay providers have been shown to effectively deliver mental health interventions (Singla et al. 2017; Sikander et al. 2019), there is reluctance in trusting the ability of task-shifted workers to carry out tasks with the same quality as a trained professionals (Kermode et al. 2021). A recent meta-analysis indicates that they may not achieve the same level of effectiveness as professionals (Venturo-Conerly et al. 2023). However, it is crucial to understand that peer-led interventions are not intended to replace those provided by trained professionals, but rather to serve as a scalable, accessible, and affordable (Singla et al. 2021) alternative to address the mental health treatment gap in low-resource settings with insufficient professional human resources. Therefore, despite these existing challenges, peer-led interventions remain a promising public health strategy for improving youth mental health in LMICs.

### Limitations

This review is not without limitations. It is possible that eligible articles were left out by our search method, either due to database selection, applied inclusion and exclusion criteria or missed search terms. Moreover, many of the studies included in this reviews are pilot studies or cross-sectional studies without a comparative group. At least three studies cited a small sample size (Mathias et al. 2018; Osborn et al. 2020a) and inadequate power (Balaji et al. 2011), which limits generalisability. There is also a potential publication bias where published literature may over-represent positive outcomes. This, along with the absence of a formal quality assessment of the included studies, can prevent a more accurate appraisal of the value of reported results to the field. Furthermore, this scoping review aimed to include all LMICs, which are inherently diverse in terms of cultural contexts, health systems, and social landscapes, potentially making generalisations among them inaccurate. A fourth limitation arises from incomplete information on the characteristics of certain interventions, where several studies offer a limited description of their intervention components. Combined with the heterogeneity of the studies included, this constituted an obstacle in comparing these studies. While the authors have attempted to minimise this by extracting information from available study protocols, the adoption of standardised presentation and evaluation for such peer-led youth mental health interventions will be helpful to support future meta-analyses and improve the comparability of study results.

## Conclusions

This scoping review highlights the breadth of peer-led interventions targeting youth mental health in LMICs, shedding light on their unique mechanisms of promoting mental health with lived experience, camaraderie and reciprocity. Future research should expand to include the perspectives of key stakeholders – peer-leaders, research teams, and regulatory bodies – focusing on factors including fidelity, feasibility and acceptability to enhance implementation insights. While peer-led interventions still rely on adult professionals support; they represent a valuable, scalable and practical strategy to bridge the human resource gap in youth mental health across LMICs.

**Open peer review.** To view the open peer review materials for this article, please visit http://doi.org/10.1017/gmh.2024.149.

**Supplementary material.** The supplementary material for this article can be found at http://doi.org/10.1017/gmh.2024.149.

**Data availability statement.** Additional review data will be shared upon request by inquiry to the corresponding author.

**Acknowledgements.** We would like to acknowledge and thank the authors of all the studies reviewed. Thank you to Beth Blackwood for helping to develop and execute the search strategy.

**Author contribution.** DWSC, DJM, DD: refining the scope of the review, data screening, extraction, analysis, literature search, manuscript writing and review; BB: development and execution of the search strategy; PP: refining the scope of the review, manuscript review; DWSC, DJM: formatting tables and figures; all authors read and approved the final manuscript.

**Financial support.** This study was supported by Duke-NUS Medical School and its Open Access publishing agreement with Cambridge.

**Competing interest.** The authors declare that the research was conducted in the absence of any commercial or financial relationships that could be construed as a potential conflict of interest.

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
