## [Reviewer Report]

Review

1. Abstract

a. Pithy and clear however results reported in results section are quite bland and predictable while the conclusions present further results.

b. Please re-structure the abstract to keep only results in results section and to use the conclusion to really pull out what this study contributes

2. Introduction

a. Please include evidence and data to explain the move to peer led interventions (including economic and efficacy)

b. Please include further discussion around the value and contribution of lived experience to implementation of mental health programmes

c. Inclusion of studies –

d. Please clarify why this review failed to examine possible contexts and mechanisms of peer – led interventions - or consider if these are angles that you could retrospectively include – this is critical to understand and scale peer led programmes

3. Methods

a. Please clarify the logic in your inclusion criteria of all LMIC countries – recognising that these are diverse contexts across several continents

b. Please clarify whether you considered hand-searching references of included papers or any other mechanism beyond the literature search – if not, why not?

c. Could you clarify the grounds for excluding this study that was a realist evaluation of the same intervention – a quantitative study by Kermode et al (2021) which was included:

i. Mathias, K., Singh, P., Butcher, N., Grills, N., Srinivasan, V., & Kermode, M. (2019). Promoting social inclusion for young people affected by psycho-social disability in India–a realist evaluation of a pilot intervention. Global Public Health, 14(12), 1718-1732.

4. Results

a. The results could be more engaging by pulling out clearly what peer-led interventions contribute which other interventions can’t achieve. The current layout and narrative reporting says in brief – peer led interventions seem to be effective to some extent and at least don’t hurt anyone - but there are challenges.

b. The results fail to engage with some of the most exciting aspects of peer led interventions – the contribution of lived experience which makes interventions more relevant and equitable and contextually acceptable eg see these studies (Atif, 2015; Maselko et al., 2020; Moreno et al., 2020; Patel et al., 2007; Petersen et al., 2012).

c. Could the results further examine some of the processes involved in peer-delivery of interventions for youth mental health e.g. comparing training, remuneration, supervision, support, safe-guarding – these are all very relevant for this study question.

d. Please consider further extracting findings to examine ithe active mechanisms that lead to outcomes with peer-interventions. It seems very likely that they are highly effective and possibly for different reasons to interventions led by adults or non-peers but you have not made any effort to identify what it is about peer-led interventions that are effective, and rather, only suggest that they don’t hurt.

e. Did you consider examining any of the other programmatic benefits from implementation with peers e.g. using local resources, local language being used, lower costs potentially, and also the benefits to the young people acting as peers? All these are questions that are likely to be able to be answered in results and I believe further data extraction and analysis could enrich this study significantly.

5. Discussion

6. Similarly to the results above – I believe the discussion would be strengthened with a more engaged discussion on the merits and possible benefits of peer support and there are strong studies that engage with this which have not been cited.

7. Implications could be further expanded

8. Limitations – surely the heterogeneity of the contexts of all LMICs is a limiting factor?

9. Conclusions

These conclusions are adequate but not sparkling. The whole paper needs more of a rationale for what it contributes and what is important, necessary and exciting about peer-led mental health interventions for youth.

Atif, N. (2015). The acceptability of peer volunteers as delivery agents of a psychosocial intervention for perinatal depression in rural Pakistan: a qualitative study. The University of Manchester (United Kingdom).

Maselko, J., Sikander, S., Turner, E. L., Bates, L. M., Ahmad, I., Atif, N., Baranov, V., Bhalotra, S., Bibi, A., & Bibi, T. (2020). Effectiveness of a peer-delivered, psychosocial intervention on maternal depression and child development at 3 years postnatal: a cluster randomised trial in Pakistan. The Lancet Psychiatry, 7(9), 775-787.

Moreno, C., Wykes, T., Galderisi, S., Nordentoft, M., Crossley, N., Jones, N., Cannon, M., Correll, C. U., Byrne, L., Carr, S., Chen, E. Y. H., Gorwood, P., Johnson, S., Kärkkäinen, H., Krystal, J. H., Lee, J., Lieberman, J., López-Jaramillo, C., Männikkö, M., Phillips, M. R., Uchida, H., Vieta, E., Vita, A., & Arango, C. (2020, Sep). How mental health care should change as a consequence of the COVID-19 pandemic. Lancet Psychiatry, 7(9), 813-824. https://doi.org/10.1016/s2215-0366(20)30307-2

Patel, V., Flisher, A. J., Hetrick, S., & McGorry, P. (2007, Apr 14). Mental health of young people: a global public-health challenge. Lancet, 369(9569), 1302-1313. https://doi.org/10.1016/S0140-6736(07)60368-7

Petersen, I., Baillie, K., Bhana, A., Mental, H., & Poverty Research Programme, C. (2012, Jul). Understanding the benefits and challenges of community engagement in the development of community mental health services for common mental disorders: lessons from a case study in a rural South African subdistrict site. Transcult Psychiatry, 49(3-4), 418-437. https://doi.org/10.1177/1363461512448375

---

## [Reviewer Report]

This scoping review undertakes an important public mental health topic and explores key areas around peer-led task-shifted interventions in LMIC settings. Though not entirely sure how China falls under LMIC? Might be useful to have some clarification from authors.

Key comments and suggestions for the authors to consider are as follows

Introduction

It might be helpful to state the key set of research questions that were explored. The outcomes of course are listed which include effectiveness, common practices and challenges that were reported in the review. This will also help establish the gap more articulately that the authors aimed at filling.

Methods

As above, it will be useful to have the list of question/research questions the authors aimed at answering in the scoping review. This should be followed by the operational definitions of the outcomes of interest. For example authors do define what is a Peer. However, what was not clear what was meant by “common practice” as a construct or outcome of the the review as well as what meant by “challenges”.

The authors should also list as an appendix (if not in the methods) their key search terms. Or at least the key terms/words. It is not a systematic review, so a search strategy is not given, but for the sake of reproducibility, some search terms and key words should be listed.

It was not very clear what were the key reasons of excluding 3356 studies; some information will help establish the reason to include the 48 studies for assessing their eligibility for the review.

Lastly, some elaboration on how the qualitative studies were reviewed and processed will be useful to have.

Results

Since the authors state that they will do only a descriptive analysis, adding some basic stats when stating results of clinical effectiveness/efficacy in the text as well. The authors do put this information in the table/end column of table #1.

Elaborating the differences between the three types of peer-led interventions, their place of delivery, format (group or individual), average duration of training and supervision processes will hugely add to the content of the review. Since these are all the implementation aspects of these peer-led interventions and any descriptive information and followed by challenges will add to the quality of the review.

Authors add “benefit” as an outcome that comes nearly at the end of the paper. What was the definition of benefits? Was this other than the clinical effectiveness? This was not clear.

Discussion

Authors should add a paragraph or a couple of key next/future focus of research and/or recommendation on reporting of peer-led trials and pilot studies. What is lacking in the reporting of such studies was highlighted in the discussion section, adding some recommendations might add value.

Some general comments (editorial in nature)

Use of terms consistently will be a great value. Eg Peers are referred to by as peer supporters, peer leaders etc. Or effectiveness is then interchanged with efficacy.

Authors use Peer Leaders. It will be good to understand if this is their term or a term used in the literature. This is a far less know term and less used as well in the global mental health and global health literature. Might be worth providing some more elaboration.

---

## [Reviewer Report]

This scoping review explores the effectiveness, common practices, and challenges of peer-led mental health interventions for youth aged 10-24 in low- and middle-income countries (LMICs). The review includes 14 studies, categorized into peer-education, peer-led psychotherapy, and peer-support interventions, all of which demonstrated improvements in mental health outcomes. The findings highlight the versatility and effectiveness of peer-led interventions, driven by camaraderie, mutual respect, and reduced stigma. However, challenges such as the need for adequate training, supervision, cultural appropriateness, and institutional support are critical to ensuring the sustainability of these interventions. The review underscores the potential of peer-led models as a valuable strategy for addressing youth mental health in LMICs and advocates for their broader implementation by policymakers. Given the potential of peer-led interventions in closing the treatment gap, this systematic review is highly timely, and I recommend it for publication.

Nonetheless, the review could benefit from the following revisions:

Major Revisions:

1. Incorporation of Recent Literature: The literature review would benefit from incorporating the most recent meta-analysis of the effectiveness of youth psychotherapy interventions in LMICs (https://doi.org/10.1016/j.jaac.2022.12.005). This will help position this review within the broader context of recent advancements in the field and highlight how it expands upon previous research.

2. Timeliness of the Search: Given the rapidly evolving nature of peer-led interventions, I recommend updating the search strategy to include more recent studies. Specifically, the following relevant studies should be considered:

o Venturo-Conerly et al., 2024 (https://doi.org/10.1016/j.jaac.2024.04.015)

o Venturo-Conerly et al., 2022 (https://doi.org/10.1016/j.brat.2022.104040)

o Osborn et al., 2023 (https://doi.org/10.1016/j.eclinm.2023.102288) Including these studies will ensure that the review is comprehensive and reflects the most current evidence.

3. Enhanced Discussion on Key Issues: To provide more robust guidance for future research and implementation, the discussion section should be expanded to include a deeper exploration of the following areas:

o Training and Supervision of Peer Providers: This section could be extended to include or link to various protocols for the training and supervision of lay-providers. For instance, Venturo-Conerly et al. (2022) provides a relevant protocol (https://doi.org/10.1016/j.cbpra.2021.03.004).

o Sustainability of Peer-Led Interventions: The sustainability of these interventions depends on their integration within existing systems of care. It would be beneficial to discuss how peer-led interventions can be embedded in tiered-caregiving models, such as the approach used by Osborn et al. (2020, 2021), where clinical supervisors handle elevated symptoms and emergencies. This will help clarify that peer-providers do not operate in isolation.

o Cross-Cultural Considerations and Generalizability: While cross-cultural considerations are mentioned, the discussion could be expanded to address the generalizability of peer-led interventions across different LMICs. Additionally, the manuscript could discuss the need for culturally and contextually sensitive protocols for handling clinical emergencies during service delivery by peer-providers (e.g., https://doi.org/10.1016/j.jaac.2022.02.005).

Minor Revisions:

• Discussion of Scoping Review Methodology Limitations: The manuscript could benefit from a more detailed discussion of the limitations specific to the scoping review methodology, such as potential bias in study selection or data extraction. The absence of a formal quality assessment of the included studies and the handling of heterogeneity should also be highlighted as limitations.

• Critical Analysis of Study Methodologies: The review provides a broad overview but could delve deeper into the critical analysis of the methodologies of the included studies, particularly concerning their applicability and generalizability to other LMIC contexts.

• Call for Standardization in Evaluations: To address the challenge of not performing a meta-analysis, the manuscript could more explicitly advocate for adopting standardized evaluation approaches for peer-led youth mental health interventions. This would support future meta-analyses and improve the comparability of study results.

---

## [Reviewer Report]

Congratulations to the authors for a thoughtful study which has engaged well with reviewer feedback. This represents an important contribution to the value of peer-led interventions for mental health. I enjoyed re-reading it.

Thankyou

---

## [Editor Report]

Dear Author,

Your revised manuscript: ‘A scoping review on peer-led interventions to improve youth mental health in low- and middle-income countries’ has now been reviewed.